# Curriculum Learning for Graph Neural Networks: Which Edges Should We Learn First

**Zheng Zhang**[†]  **Junxiang Wang**[◇]  **Liang Zhao**[†]
†Emory University, Atlanta, GA  ◇NEC Labs America, Princeton, NJ
{zheng.zhang,liang.zhao}@emory.edu
{junxiang.wang}@alumni.emory.edu

## Abstract

Graph Neural Networks (GNNs) have achieved great success in representing data with dependencies by recursively propagating and aggregating messages along the edges. However, edges in real-world graphs often have varying degrees of difficulty, and some edges may even be noisy to the downstream tasks. Therefore, existing GNNs may lead to suboptimal learned representations because they usually treat every edge in the graph equally. On the other hand, Curriculum Learning (CL), which mimics the human learning principle of learning data samples in a meaningful order, has been shown to be effective in improving the generalization ability and robustness of representation learners by gradually proceeding from easy to more difficult samples during training. Unfortunately, existing CL strategies are designed for independent data samples and cannot trivially generalize to handle data dependencies. To address these issues, we propose a novel CL strategy to gradually incorporate more edges into training according to their difficulty from easy to hard, where the degree of difficulty is measured by how well the edges are expected given the model training status. We demonstrate the strength of our proposed method in improving the generalization ability and robustness of learned representations through extensive experiments on nine synthetic datasets and nine real-world datasets. The code for our proposed method is available at https://github.com/rollingstonezz/Curriculum_learning_for_GNNs.

## 1 Introduction

Inspired by cognitive science studies [8, 33] that humans can benefit from the sequence of learning basic (easy) concepts first and advanced (hard) concepts later, curriculum learning (CL) [2] suggests training a machine learning model with easy data samples first and then gradually introducing more hard samples into the model according to a designed pace, where the difficulty of samples can usually be measured by their training loss [25]. Many previous studies have shown that this easy-to-hard learning strategy can effectively improve the generalization ability of the model [2, 19, 15, 11, 35, 46], and some studies [19, 15, 11] have shown that CL strategies can also increase the robustness of the learned model against noisy training samples. An intuitive explanation is that in CL settings noisy data samples correspond to harder samples, and CL learner spends less time with the harder (noisy) samples to achieve better generalization performance and robustness.

Although CL strategies have achieved great success in many fields such as computer vision and natural language processing, existing methods are designed for independent data (such as images) while designing effective CL methods for data with dependencies has been largely underexplored. For example, in a citation network, two researchers with highly related research topics (e.g. machine learning and data mining) are more likely to collaborate with each other, while the reason behind a collaboration of two researchers with less related research topics (e.g. computer architecture and social science) might be more difficult to understand. Prediction on one sample impacts that of another, forming a graph structure that encompasses all samples connected by their dependencies. There are

37th Conference on Neural Information Processing Systems (NeurIPS 2023).

many machine learning techniques for such graph-structured data, ranging from traditional models like conditional random field [36], graph kernels [37], to modern deep models like GNNs [29, 30, 52, 38, 49, 12, 53, 42]. However, traditional CL strategies are not designed to handle the curriculum of the dependencies between nodes in graph data, which are insufficient. Handling graph-structured data require not only considering the difficulty in individual samples, but also the difficulty of their dependencies to determine how to gradually composite correlated samples for learning.

As previous CL strategies indicated that an easy-to-hard learning sequence on data samples can improve the generalization and robustness performance, an intuitive question is whether a similar strategy on data dependencies that iteratively involves easy-to-hard edges in learning can also benefit. Unfortunately, there exists no trivial way to directly generalize existing CL strategies on independent data to handle data dependencies due to several unique challenges: (1) **Difficulty in quantifying edge selection criteria**. Existing CL studies on independent data often use supervised computable metrics (e.g. training loss) to quantify sample difficulty, but how to quantify the difficulties of understanding the dependencies between data samples which has no supervision is challenging. (2) **Difficulty in designing an appropriate curriculum to gradually involve edges**. Similar to the human learning process, the model should ideally retain a certain degree of freedom to adjust the pacing of including edges according to its own learning status. As existing CL methods for graph data typically use fixed pacing function to involve samples, they can not provide this flexibility. Designing an adaptive pacing function for handling graph data is difficult since it requires joint optimization of both supervised learning tasks on nodes and the number of chosen edges. (3) **Difficulty in ensuring convergence and a numerical steady process for CL in graphs**. Discrete changes in the number of edges can cause drift in the optimal model parameters between training iterations. How to guarantee a numerically stable learning process for CL on edges is challenging.

In order to address the aforementioned challenges, in this paper, we propose a novel CL algorithm named **R**elational **C**urriculum **L**earning (**RCL**) to improve the generalization ability and robustness of representation learners on data with dependencies. To address the first challenge, we propose an approach to select the edges by quantifying their corresponding difficulties in a self-supervised learning manner. Specifically, for each training iteration, we choose $K$ easiest edges whose corresponding relations are most well-expected by the current model. Second, to design an appropriate learning pace for gradually involving more edges in training, we present the learning process as a concise optimization model, which automatically lets the model gradually increase the number $K$ to involve more edges in training according to its own status. Third, to ensure convergence of optimizing the model, we propose an alternative optimization algorithm with a theoretical convergence guarantee and an edge reweighting scheme to smooth the graph structure transition. Finally, we demonstrate the superior performance of RCL compared to state-of-the-art comparison methods through extensive experiments on both synthetic and real-world datasets.

## 2 Related Works

**Curriculum Learning (CL).**    Bengio et al.[2] pioneered the concept of Curriculum Learning (CL) within the machine learning domain, aiming to improve model performance by gradually including easy to hard samples in training the model. Self-paced learning [25] measures the difficulty of samples by their training loss, which addressed the issue in previous works that difficulties of samples are generated by prior heuristic rules. Therefore, the model can adjust the curriculum of samples according to its own training status. Following works [18, 17, 55] further proposed many supervised measurement metrics for determining curriculums, for example, the diversity of samples [17] or the consistency of model predictions [55]. Meanwhile, many empirical and theoretical studies were proposed to explain why CL could lead to generalization improvement from different perspectives. For example, studies such as MentorNet [19] and Co-teaching [15] empirically found that utilizing CL strategy can achieve better generalization performance when the given training data is noisy. [11] provided theoretical explanations on the denoising mechanism that CL learners waste less time with the noisy samples as they are considered harder samples. Some studies [2, 35, 46, 13, 24] also realized that CL can help accelerate the optimization process of non-convex objectives and improve the speed of convergence in the early stages of training.

Despite great success, most of the existing designed CL strategies are for independent data such as images, and there is little work on generalizing CL strategies to handle samples with dependencies. Few existing attempts on graph-structured data [26, 21, 28], such as [44, 5, 45, 28], simply treat

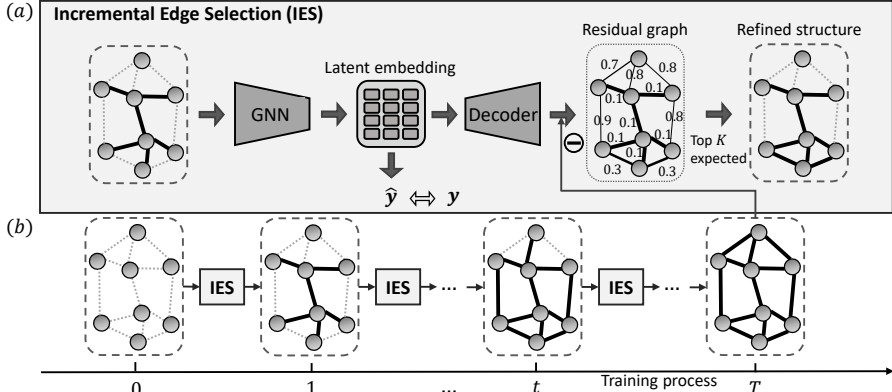

Figure 1: The overall framework of RCL. (a) The *Incremental Edge Selection* module first extracts the latent node embedding by the GNN model given the current training structure, then jointly learns the node prediction label **y** and reconstructs the input structure by a decoder. A small residual error on an edge indicates the corresponding dependency is well expected and thus can be added to the refined structure for the next iteration. (b) The iterative learning process of RCL. The model starts with an empty structure and gradually includes more edges until the training structure converges to the input structure.

nodes as independent samples and then apply CL strategies on independent data, which ignore the fundamental and unique dependency information carried by the structure in data, and thus can not well handle the correlation between data samples. Furthermore, these models are mostly based on heuristic-based sample selection strategies [5, 45, 28], which largely limit the generalizability of these methods.

**Graph structure learning.** Another stream of existing studies that are related to our work is *graph structure learning*. Recent studies have shown that GNN models are vulnerable to adversarial attacks on graph structure [7, 48]. In order to address this issue, studies in *graph structure learning* usually aim to jointly learn an optimized graph structure and corresponding graph representations. Existing works [9, 4, 20, 54, 31] typically consider the hypothesis that the intrinsic graph structure should be sparse or low rank from the original input graph by pruning "irrelevant" edges. Thus, they typically use pre-deterministic methods [7, 56, 9] to preprocess graph structure such as singular value decomposition (SVD), or dynamically remove "redundant" edges according to the downstream task performance on the current sparsified structure [4, 20, 31]. However, modifying the graph topology will inevitably lose potentially useful information lying in the removed edges. More importantly, the modified graph structure is usually optimized for maximizing the performance on the training set, which can easily lead to overfitting issues.

## 3 Preliminaries

Graph neural networks (GNNs) are a class of methods that have shown promising progress in representing structured data in which data samples are correlated with each other. Typically, the data samples are treated as nodes while their dependencies are treated as edges in the constructed graph. Formally, we denote a graph as $G = (\mathcal{V}, \mathcal{E})$, where $\mathcal{V} = \{v_1, v_2, \ldots, v_N\}$ is a set of nodes that $N = |\mathcal{V}|$ denotes the number of nodes in the graph and $\mathcal{E} \subseteq \mathcal{V} \times \mathcal{V}$ is the set of edges. We also let $\mathbf{X} \in \mathbb{R}^{N \times b}$ denote the node attribute matrix and let $\mathbf{A} \in \mathbb{R}^{N \times N}$ represent the adjacency matrix. Specifically, $A_{ij} = 1$ denotes that there is an edge connecting nodes $v_i$ and $v_j \in \mathcal{V}$, otherwise $A_{ij} = 0$. A GNN model $f$ maps the node feature matrix $\mathbf{X}$ associated with the adjacency matrix $\mathbf{A}$ to the model predictions $\hat{\mathbf{y}} = f(\mathbf{X}, \mathbf{A})$, and get the loss $L_{\text{GNN}} = L(\hat{\mathbf{y}}, \mathbf{y})$, where $L$ is the objective function and $\mathbf{y}$ is the ground-truth label of nodes. The loss on one node $v_i$ is denoted as $l_i = L(\hat{y}_i, y_i)$.

## 4 Methodology

As previous CL methods have shown that an easy-to-hard learning sequence of independent data samples can improve the generalization ability and robustness of the representation learner, the goal of this paper is to develop an effective CL method on data with dependencies, which is extremely difficult due to several unique challenges: (1) Difficulty in designing a feasible principle to select

edges by properly quantifying their difficulties. (2) Difficulty in designing an appropriate pace of curriculum to gradually involve more edges in training based on model status. (3) Difficulty in ensuring convergence and a numerical steady process for optimizing the CL model.

In order to address the above challenges, we propose a novel CL method named **R**elational **C**urriculum **L**earning (**RCL**). The sequence, which gradually includes edges from easy to hard, is called *curriculum* and learned in different grown-up stages of training. In order to address the first challenge, we propose a self-supervised module *Incremental Edge Selection (IES)*, which is shown in Figure 1(a), to select the $K$ easiest edges at each training iteration that are mostly expected by the current model. The details are elaborated in Section 4.1. To address the second challenge, we present a joint optimization framework to automatically increase the number of selected edges $K$ given its own training status. The framework is elaborated in Figure 1(b) and details can be found in Section 4.2. Finally, to ensure convergence of optimization and steady the numerical process, we propose an EM-style alternative optimization algorithm with a theoretical convergence guarantee in Section 4.2 Algorithm 1 and an edge reweighting scheme to smooth the discrete edge incrementing process in Section 4.3.

## 4.1 Incremental Edge Selection by Quantifying Difficulties of Sample Dependencies

Here we propose a novel way to select edges by first quantifying their difficulty levels. Existing works on independent data typically use supervised metrics such as training loss of samples to quantify their difficulty level, but there exists no supervised metrics on edges. To address this issue, we propose a self-supervised module *Incremental Edge Selection (IES)*. We first quantify the difficulty of edges by measuring how well the edges are expected from the currently learned embeddings of their connected nodes. Then the most well-expected edges are selected as the easiest edges for the next iteration of training. As shown in Figure 1(a), given the currently selected edges at iteration $t$, we first feed them to the GNN model to extract the latent node embeddings. Then we restore the latent node embeddings to the original graph structure through a decoder, which is called the reconstruction of the original graph structure. The residual graph $\mathbf{R}$, which is defined as the degree of mismatch between the original adjacency matrix $\mathbf{A}$ and the reconstructed adjacency matrix $\tilde{\mathbf{A}}^{(t)}$, can be considered a strong indicator for describing how well the edges are expected by the current model. Specifically, a smaller residual error indicates a higher probability of being a well-expected edge.

With the developed self-supervised method to measure the difficulties of edges, here we formulate the key learning paradigm of selecting the top $K$ easiest edges. To obtain the training adjacency matrix $\mathbf{A}^{(t)}$ that will be fed into the GNN model $f^{(t)}$, we introduce a learnable binary mask matrix $\mathbf{S}$ with each element $\mathbf{S}_{ij} \in \{0, 1\}$. Thus, the training adjacency matrix at iteration $t$ can be represented as $\mathbf{A}^{(t)} = \mathbf{S}^{(t)} \odot \mathbf{A}$. To filter out the edges with $K$ smallest residual error, we penalize the summarized residual errors over the selected edges, which can be represented as $\sum_{i,j} \mathbf{S}_{ij} \mathbf{R}_{ij}$. Therefore, the learning objective can be presented as follows:

$$\min_{\mathbf{w}} L_{\text{GNN}} + \beta \sum_{i,j} \mathbf{S}_{ij} \mathbf{R}_{ij},$$
$$s.t. \|\mathbf{S}\|_1 \geq K,$$
(1)

where the first term $L_{\text{GNN}} = L(f(\mathbf{X}, \mathbf{A}^{(t)}; \mathbf{w}), \mathbf{y})$ is the node-level predictive loss, e.g. cross-entropy loss for the node classification task. The second term $\sum_{i,j} \mathbf{S}_{ij} \mathbf{R}_{ij}$ aims at penalizing the residual errors over the edges selected by the mask matrix $\mathbf{S}$. $\beta$ is a hyperparameter to tune the balance between terms. The constraint is to guarantee only the most $K$ well-expected edges are selected.

More concretely, the value of a residual edge $\tilde{\mathbf{A}}_{ij}^{(t)} \in [0, 1]$ can be computed by a non-parametric kernel function $\kappa(\mathbf{z}_i^{(t)}, \mathbf{z}_j^{(t)})$, e.g. the inner product kernel. Then the residual error $\mathbf{R}_{ij}$ between the input structure and the reconstructed structure can be defined as $\left\|\tilde{\mathbf{A}}_{ij}^{(t)} - \mathbf{A}_{ij}\right\|$, where $\|\cdot\|$ is commonly chosen to be the squared $\ell_2$-norm.

## 4.2 Automatically Control the Pace of Increasing Edges

In order to dynamically include more edges into training, an intuitive way is to iteratively increase the value of $K$ in Equation 1 to allow more edges to be selected. However, it is difficult to determine an appropriate value of $K$ with respect to the training status of the model. Besides, directly solving

---
**Algorithm 1** Alternating Minimization Algorithm for Optimizing Equation 2
---
**Input:** Node features $\mathbf{X}$, adjacency matrix $\mathbf{A}$, stepsize $\mu$ and hyperparameter $\gamma$
**Output:** Parameters $\mathbf{w}$ of GNN model $f$

1: Initialize $\mathbf{w}^{(0)}, \mathbf{S}^{(0)}, \lambda$
2: **while** Not converged **do**
3:    $\mathbf{w}^{(t)} = \arg\min_{\mathbf{w}} L(f(\mathbf{X}, \mathbf{A}^{(t-1)}; \mathbf{w}), \mathbf{y}) + \beta \sum_{i,j} \mathbf{S}_{ij} \left\| \tilde{\mathbf{A}}_{ij}^{(t-1)} - \mathbf{A}_{ij} \right\| + \frac{\gamma}{2} \left\| \mathbf{w} - \mathbf{w}^{(t-1)} \right\|$
4:    Given $\mathbf{w}^{(t)}$, extract latent nodes embedding $\mathbf{Z}^{(t)}$ from GNN model $f$
5:    Calculate reconstructed structure $\tilde{\mathbf{A}}_{ij}^{(t)} = \kappa(\mathbf{z}_i^{(t)}, \mathbf{z}_j^{(t)})$ for all pairs of $i, j$
6:    $\mathbf{S}^{(t)} = \arg\min_{\mathbf{S}} \beta \sum_{i,j} \mathbf{S}_{ij} \left\| \mathbf{A}_{ij} - \tilde{\mathbf{A}}_{ij}^{(t)} \right\| + g(\mathbf{S}; \lambda) + \frac{\gamma}{2} \left\| \mathbf{S} - \mathbf{S}^{(t-1)} \right\|$
7:    Compute $\mathbf{A}^{(t)} = \mathbf{S}^{(t)} \odot \mathbf{A}$
8:    **if** $\mathbf{A}^{(t)} \neq \mathbf{A}$ **then**
9:        Increase $\lambda$ by stepsize $\mu$
10:    **end if**
11: **end while**
---

Equation 1 is difficult since $\mathbf{S}$ is a binary matrix where each element $\mathbf{S}_{ij} \in \{0, 1\}$, optimizing $\mathbf{S}$ would require solving a discrete constraint program at each iteration. To address this issue, we first relax the problem into continuous optimization so that each $\mathbf{S}_{ij}$ can be allowed to take any value in the interval $[0, 1]$. Note that the inequality $||\mathbf{S}||_1 \geq K$ in Eqn. 1 is equivalent to the equality $||\mathbf{S}||_1 = K$. This is because the second term in the loss function would always encourage fewer selected edges by the mask matrix $\mathbf{S}$, as all values in the residual error matrix $\mathbf{R}$ and mask matrix $\mathbf{S}$ are nonnegative. Given this, we can incorporate the equality constraint as a Lagrange multiplier and rewrite the loss function as $\mathcal{L} = L_{GNN} + \beta \sum_{i,j} \mathbf{S}_{ij} \mathbf{R}_{ij} - \lambda(||\mathbf{S}||_1 - K)$. Considering that $K$ remains constant, the optimization of the loss function can be equivalently framed by substituting the given constraint with a regularization term denoted as $g(\mathbf{S}; \lambda)$. As such, the overall loss function can be reformulated as:

$$\min_{\mathbf{w}, \mathbf{S}} L_{\text{GNN}} + \beta \sum_{i,j} \mathbf{S}_{ij} \mathbf{R}_{ij} + g(\mathbf{S}; \lambda), \tag{2}$$

where $g(\mathbf{S}; \lambda) = \lambda \|\mathbf{S} - \mathbf{A}\|$ and $\|\cdot\|$ is commonly chosen to be the squared $\ell_2$-norm. Since the training adjacency matrix $\mathbf{A}^{(t)} = \mathbf{S}^{(t)} \odot \mathbf{A}$, as $\lambda \to \infty$, more edges in the input structure are included until the training adjacency matrix $\mathbf{A}^{(t)}$ converges to the input adjacency matrix $\mathbf{A}$. Specifically, the regularization term $g(\mathbf{S}; \lambda)$ controls the learning scheme by the *age parameter* $\lambda$, where $\lambda = \lambda(t)$ grows with the number of iterations. By monotonously increasing the value of $\lambda$, the regularization term $g(\mathbf{S}; \lambda)$ will push the mask matrix gradually converge to the input adjacency matrix $\mathbf{A}$, resulting in more edges automatically involved in the training structure.

**Optimization of learning objective.** In optimizing the objective function in Equation 2, we need to jointly optimize parameter $\mathbf{w}$ for GNN model $f$ and the mask matrix $\mathbf{S}$. To tackle this, we introduce an EM-style optimization scheme (detailed in Algorithm 1) that iteratively updates both. The algorithm uses the node feature matrix $\mathbf{X}$, the original adjacency matrix $\mathbf{A}$, a step size $\mu$ to control the age parameter $\lambda$ increase rate, and a hyperparameter $\gamma$ for regularization adjustments. Post initialization of $\mathbf{w}$ and $\mathbf{S}$, it alternates between: optimizing GNN model $f$ (Step 3), extracting latent node embeddings and reconstructing the adjacency matrix (Steps 4 & 5), refining the mask matrix using the reconstructed matrix and regularization, and results in more edges are gradually involved (Step 6), updating the training adjacency matrix (Step 7), and incrementing $\lambda$ when the training matrix $\mathbf{A}^{(t)}$ differs from input matrix $\mathbf{A}$, incorporating more edges in the next iteration.

**Theorem 4.1.** *We have the following convergence guarantees for Algorithm 1:*
• *Avoidance of Saddle Points. If the second derivatives of $L(f(\mathbf{X}, \mathbf{A}^{(t)}; \mathbf{w}), \mathbf{y})$ and $g(\mathbf{S}; \lambda)$ are continuous, then for sufficiently large $\gamma$, any bounded sequence $(\mathbf{w}^{(t)}, \mathbf{S}^{(t)})$ generated by Algorithm 1 with random initializations will not converge to a strict saddle point of $F$ almost surely.*
• *Second Order Convergence. If the second derivatives of $L(f(\mathbf{X}, \mathbf{A}^{(t)}; \mathbf{w}), \mathbf{y})$ and $g(\mathbf{S}; \lambda)$ are continuous, and $L(f(\mathbf{X}, \mathbf{A}^{(t)}; \mathbf{w}), \mathbf{y})$ and $g(\mathbf{S}; \lambda)$ satisfy the Kurdyka-Łojasiewicz (KL) property [41], then for sufficiently large $\gamma$, any bounded sequence $(\mathbf{w}^{(t)}, \mathbf{S}^{(t)})$ generated by Algorithm 1 with random initialization will almost surely converge to a second-order stationary point of $F$.*

The detailed proof can be found in Appendix B.

### 4.3 Smooth Structure Transition by Edge Reweighting

Note that in the Algorithm 1, the optimization process requires iteratively updating the parameters $\mathbf{w}$ of the GNN model $f$ and current adjacency matrix $\mathbf{A}^{(t)}$, where $\mathbf{A}^{(t)}$ varies discretely between iterations. However, GNN models mostly work in a message-passing fashion, which computes node representations by iteratively aggregating information along edges from neighboring nodes. Discretely modifying the number of edges will result in a great drift of the optimal model parameters between iterations. In Appendix Figure , we demonstrate that a shift in the optimal parameters of the GNN results in a spike in the training loss. Therefore, it can increase the difficulty of finding optimal parameters and even hurt the generalization ability of the model in some cases. Besides the numerical problem caused by discretely increasing the number of edges, another issue raised by the CL strategy in Section 4.1 is the trustworthiness of the estimated edge difficulty, which is inferred by the residual error on the edges. Although the residual error can reflect how well edges are expected in the ideal case, the quality of the learned latent node embeddings may affect the validity of this metric and compromise the quality of the designed curriculum by the CL strategy.

To address both issues, we propose a novel edge reweighting scheme to (1) smooth the transition of the training structure between iterations, and (2) reduce the weight of edges that connect nodes with low-confidence latent embeddings. Formally, we use a smoothed version of structure $\bar{\mathbf{A}}^{(t)}$ to substitute $\mathbf{A}^{(t)}$ for training the GNN model $f$ in step 3 of Algorithm 1, where the mapping from $\mathbf{A}^{(t)}$ to $\bar{\mathbf{A}}^{(t)}$ can be represented as:

$$\bar{\mathbf{A}}_{ij}^{(t)} = \pi_{ij}^{(t)} \mathbf{A}_{ij}^{(t)}, \tag{3}$$

where $\pi_{ij}^{(t)}$ is the weight imposed on edge $e_{ij}$ at iteration $t$. $\pi_{ij}^{(t)}$ is calculated by considering the counted occurrences of edge $e_{ij}$ until the iteration $t$ and the confidence of the latent embedding for the connected pair of nodes $v_i$ and $v_j$:

$$\pi_{ij}^{(t)} = \psi(e_{ij})\rho(v_i)\rho(v_j), \tag{4}$$

where $\psi$ is a function that reflects the number of edge occurrences and $\rho$ is a function to reflect the degree of confidence for the learned latent node embedding. The details of these two functions are described as follow.

**Smooth the transition of the training structure between iterations.** In order to obtain a smooth transition of the training structure between iterations, we take the learned curriculum of selected edges into consideration. Formally, we model $\psi$ by a smooth function of the edge selected occurrences compared to the model iteration occurrences before the current iteration:

$$\psi(e_{ij}) = t(e_{ij})/t, \tag{5}$$

where $t$ is the number of current iterations and $t(e_{ij})$ represents the counting number of selecting edge $e_{ij}$. Therefore, we transform the original discretely changing training structure into a smoothly changing one by taking the historical edge selection curriculum into consideration.

**Reduce the influence of nodes with low confidence latent embeddings.** As introduced in our Algorithm 1 line 6, the estimated structure $\tilde{A}$ is inferred from the latent embedding $\mathbf{Z}$, which is extracted from the trained GNN model $f$. Such estimated latent embedding may possibly differ from the true underlying embedding, which results in the inaccurately reconstructed structure around the node. In order to alleviate this issue, we model the function $\rho$ by the training loss on nodes, which indicates the confidence of their learned latent embeddings. This idea is similar to previous CL strategies on inferring the difficulty of data samples by their supervised training loss. Specifically, a larger training loss indicates a low confident latent node embedding. Mathematically, the weights $\rho(v_i)$ on node $v_i$ can be represented as a distribution of their training loss:

$$\rho(v_i) \sim e^{-l_i} \tag{6}$$

where $l_i$ is the training loss on node $v_i$. Therefore, a node with a larger training loss will result in a smaller value of $\rho(v_i)$, which reduces the weight of its connecting edges.

## 5 Experiments

In this section, the experimental settings are introduced first in Section 5.1, then the performance of the proposed method on both synthetic and real-world datasets are presented in Section 5.2. We further present the robustness test on our CL method against topological structure noise in Section 5.3.

Table 1: Node classification accuracy on synthetic datasets (%). The best-performing method on each backbone GNN model is highlighted in bold, while the second-best method is underlined. In situations where RCL's performance is not strictly the best among all methods, we can see that almost all methods can achieve a near-perfect performance and RCL is still close to the best methods.

| Homo ratio | 0.1 | 0.2 | 0.3 | 0.4 | 0.5 | 0.6 | 0.7 | 0.8 | 0.9 |
|---|---|---|---|---|---|---|---|---|---|
| GCN | 50.84±1.03 | 56.50±0.50 | 65.17±0.48 | 77.94±0.54 | 87.15±0.44 | 93.27±0.24 | 97.48±0.25 | 99.10±0.17 | 99.93±0.03 |
| GNNSVD | 54.96±0.76 | 58.45±0.56 | 63.06±0.63 | 70.23±0.61 | 80.51±0.41 | 85.02±0.46 | 90.31±0.27 | 94.23±0.22 | 96.74±0.23 |
| ProGNN | 47.87±0.87 | 54.59±0.55 | 65.39±0.44 | 76.96±0.49 | 87.76±0.51 | 93.16±0.34 | 97.60±0.31 | 99.04±0.19 | **99.94±0.03** |
| NeuralSparse | 51.42±1.35 | 57.99±0.69 | 65.10±0.43 | 75.37±0.34 | 87.40±0.29 | 93.54±0.28 | 97.16±0.15 | 99.01±0.22 | 99.83±0.07 |
| PTDNet | 48.21±1.98 | 55.52±2.82 | 65.82±0.94 | 79.37±0.45 | 89.17±0.39 | 94.19±0.18 | 98.61±0.12 | 99.51±0.09 | 99.81±0.05 |
| CLNodes | 50.37±0.73 | 56.64±0.56 | 65.04±0.66 | 77.52±0.48 | 86.85±0.44 | 93.10±0.47 | 97.34±0.25 | 99.02±0.18 | 99.88±0.04 |
| RCL | **57.57±0.43** | **62.06±0.28** | **73.98±0.55** | **84.54±0.75** | **92.69±0.09** | **97.42±0.17** | **99.62±0.05** | **99.89±0.02** | 99.93±0.06 |
| GIN | 48.33±1.89 | 53.62±1.39 | 64.08±0.99 | 77.55±1.10 | 85.31±0.75 | 90.57±0.36 | 97.82±0.18 | 99.59±0.11 | 99.91±0.02 |
| GNNSVD | 43.21±1.60 | 45.68±1.66 | 54.90±1.16 | 68.29±0.79 | 79.76±0.52 | 85.63±0.44 | 93.65±0.39 | 97.22±0.17 | 98.94±0.17 |
| ProGNN | 45.76±1.40 | 52.96±1.01 | 64.12±1.07 | 76.95±0.87 | 85.13±0.71 | 89.96±0.55 | 96.54±0.48 | 99.51±0.12 | 99.78±0.05 |
| NeuralSparse | 50.23±2.05 | 54.12±1.52 | 62.81±0.75 | 76.98±1.17 | 85.14±0.94 | 92.57±0.44 | 98.02±0.20 | 99.61±0.12 | 99.91±0.05 |
| PTDNet | 53.23±2.76 | 56.12±2.03 | 65.81±1.38 | 77.81±1.02 | 86.14±0.65 | 93.21±0.74 | 97.08±0.41 | 99.51±0.18 | 99.91±0.03 |
| CLNodes | 45.36±1.42 | 51.10±1.15 | 62.53±0.88 | 75.83±1.07 | 87.76±0.90 | 94.25±0.44 | 98.30±0.26 | 99.60±0.09 | **99.92±0.03** |
| RCL | **57.63±0.66** | **62.08±1.17** | **71.02±0.61** | **80.61±0.69** | **88.62±0.43** | **94.88±0.36** | 98.19±0.19 | 99.32±0.08 | 99.89±0.04 |
| GraphSAGE | 62.57±0.55 | 67.33±0.64 | 71.06±0.74 | 80.88±0.54 | 85.88±0.51 | 91.42±0.37 | 95.26±0.33 | 97.78±0.16 | 99.52±0.13 |
| GNNSVD | 64.42±0.80 | 65.71±0.39 | 67.12±0.58 | 68.47±0.50 | 77.70±0.65 | 82.86±0.50 | 87.81±0.71 | 91.61±0.55 | 95.01±0.50 |
| ProGNN | 58.57±2.09 | 66.75±0.91 | 72.14±0.64 | 81.27±0.44 | 86.89±0.47 | 92.10±0.39 | 95.21±0.30 | 97.51±0.23 | 99.50±0.11 |
| NeuralSparse | 61.70±0.77 | 66.65±0.66 | 70.60±0.79 | 79.65±0.45 | 84.19±0.91 | 91.31±0.54 | 94.86±0.53 | 97.16±0.23 | 99.55±0.19 |
| PTDNet | 65.72±1.08 | 69.25±0.92 | 72.60±0.77 | 79.65±0.45 | 86.54±0.56 | 91.79±0.53 | 96.10±0.58 | 97.98±0.13 | **99.78±0.08** |
| CLNodes | **69.41±0.66** | 70.83±0.58 | 75.51±0.36 | 82.65±0.43 | 87.08±0.56 | 91.58±0.41 | 95.91±0.38 | 98.33±0.26 | 99.57±0.14 |
| RCL | 68.03±0.37 | **71.39±0.51** | **76.99±0.99** | **83.76±0.55** | **88.24±0.30** | 93.34±0.56 | **97.66±0.52** | **98.86±0.28** | 99.64±0.08 |

We verify the effectiveness of framework components through ablation studies in Section 5.4. Intuitive visualizations of the edge selection curriculum are shown in Section 5.5. In addition, we measure the parameter sensitivity in Appendix A.2 and running time analysis in Appendix A.5 due to the space limit.

## 5.1 Experimental Settings

**Synthetic datasets.** To evaluate the effectiveness of our proposed method on datasets with ground-truth difficulty labels on edges, we follow previous studies [22, 1] to generate a set of synthetic datasets, where the formation probability of an edge is designed to reflect its likelihood to positively contribute to the node classification job, which indicates its ground-truth difficulty level. Specifically, the nodes in a generated graph are divided into 10 equally sized node classes $1, 2, \ldots, 10$, and the node features are sampled from overlapping multi-Gaussian distributions. Each generated graph is associated with a *homophily coefficient (homo)* which indicates the probability of a node forming an edge to another node with the same label. For the rest edges that are formed between nodes with different labels, the probability of forming an edge is inversely proportional to the distances between their labels. Nodes with close classes are more likely to be connected since the formation probability decreases with the distance of the node label, and connections from nodes with close classes can increase the likelihood of accurately classifying a node due to the homophily property of the designed node classification task. Therefore, an edge with a high formation probability indicates a higher chance to positively contribute to the node classification task because it connects a node with a close class, and thus can be considered an easy edge. We vary the value of *homo* to generate nine graphs in total. More details and visualization about the synthetic dataset can be found in Appendix A.1.

**Real-world datasets.** To further evaluate the performance of our proposed method in real-world scenarios, nine benchmark real-world attributed network datasets, including four citation network datasets Cora, Citeseer, Pubmed [51] and ogbn-arxiv [16], two coauthor network datasets CS and Physics [34], two Amazon co-purchase network datasets Photo and Computers [34], and one protein interaction network ogbn-proteins [16]. We follow the data splits from [3] on citation networks and use a 5-fold cross-validation setting on coauthor and Amazon co-purchase networks. All datasets are publicly available from Pytorch-geometric library [10] and Open Graph Benchmark (OGB) [16], where basic statistics are reported in Table 2.

**Comparison methods.** We incorporate three commonly used GNN models, including GCN [23], GraphSAGE [14], and GIN [50], as the baseline model and also the backbone model for RCL. In addition to evaluating our proposed method against the baseline GNNs, we further leverage two categories of state-of-the-art comparison methods in the experiments: (1) We incorporate four graph structure learning methods GNNSVD [9], ProGNN [20], NeuralSparse [54], and PTDNet [31]; (2) We further compare with a curriculum learning method named CLNode [45] which gradually select nodes in the order of the difficulties defined by a heuristic-based strategy. More details about comparison methods can be found in Appendix A.1.

Table 2: Node classification results on real-world datasets (%). The best-performing method on each backbone is highlighted in bold and second-best is underlined. (OOM) shorts for out-of-memory.

| | Cora | Citeseer | Pubmed | CS | Physics | Photo | Computers | ogbn-arxiv | ogbn-proteins |
|---|---|---|---|---|---|---|---|---|---|
| # nodes | 2,708 | 3,327 | 19,717 | 18,333 | 34,493 | 7,650 | 13,752 | 169,343 | 132,534 |
| # edges | 10,556 | 9,104 | 88,648 | 163,788 | 495,924 | 238,162 | 491,722 | 1,166,243 | 39,561,252 |
| # features | 1,433 | 3,703 | 500 | 6,805 | 8,415 | 745 | 767 | 100 | 8 |
| GCN | 85.74±0.42 | 78.93±0.32 | 87.91±0.09 | 93.03±0.32 | 96.55±0.15 | 93.25±0.70 | 88.09±0.40 | 71.74±0.29 | 72.51±0.35 |
| GNNSVD | 83.24±1.03 | 74.80±0.87 | 88.81±0.38 | 93.79±0.11 | 96.11±0.13 | 89.63±0.73 | 86.49±0.77 | 67.44±0.51 | 66.92±0.64 |
| ProGNN | 85.66±0.61 | 74.78±0.55 | 87.22±0.33 | 94.04±0.19 | 96.75±0.26 | 92.07±0.67 | 88.72±0.59 | (OOM) | (OOM) |
| NeuralSparse | 85.95±0.98 | 76.24±0.48 | 86.83±0.40 | 92.31±0.47 | 95.56±0.30 | 90.57±0.90 | 88.62±0.83 | (OOM) | (OOM) |
| PTDNet | 83.84±0.95 | 77.54±0.42 | 87.89±0.08 | 93.60±0.43 | 96.56±0.09 | 88.92±0.87 | 87.52±0.70 | (OOM) | (OOM) |
| CLNode | 85.67±0.33 | 78.99±0.57 | 89.50±0.28 | 93.83±0.24 | 95.76±0.16 | 93.39±0.83 | 89.28±0.38 | 70.95±0.18 | 71.40±0.32 |
| RCL | 87.15±0.44 | 79.79±0.55 | 89.79±0.12 | 94.66±0.32 | 97.02±0.23 | 94.41±0.76 | 90.23±0.23 | 74.08±0.33 | 75.19±0.26 |
| GIN | 84.43±0.65 | 74.87±0.20 | 85.72±0.40 | 91.48±0.36 | 95.62±0.30 | 93.02±0.91 | 86.94±1.58 | 69.26±0.34 | 74.51±0.32 |
| GNNSVD | 82.23±0.65 | 72.11±0.70 | 88.31±0.15 | 91.40±0.87 | 95.30±0.29 | 89.49±1.11 | 82.66±2.26 | 67.79±0.41 | 70.65±0.53 |
| ProGNN | 85.02±0.41 | 78.12±0.93 | 87.82±0.51 | (OOM) | 95.23±0.67 | 83.54±1.48 | (OOM) | (OOM) |
| NeuralSparse | 84.92±0.58 | 75.44±0.87 | 86.11±0.49 | 89.66±0.82 | 95.05±0.57 | 93.28±0.83 | 87.22±0.54 | (OOM) | (OOM) |
| PTDNet | 83.02±1.01 | 75.00±0.74 | 88.04±0.29 | 91.01±0.21 | 95.57±0.40 | 90.70±0.76 | 87.08±0.65 | (OOM) | (OOM) |
| CLNode | 83.52±0.77 | 75.82±0.58 | 86.92±0.61 | 91.71±0.41 | 95.75±0.46 | 92.78±0.90 | 85.93±1.53 | 70.58±0.17 | 73.97±0.31 |
| RCL | 86.64±0.39 | 77.60±0.18 | 89.17±0.29 | 93.92±0.27 | 96.75±0.17 | 93.88±0.51 | 89.76±0.19 | 72.55±0.15 | 78.76±0.22 |
| GraphSAGE | 86.22±0.27 | 77.27±0.23 | 88.50±0.16 | 94.22±0.18 | 96.26±0.34 | 93.82±0.51 | 88.62±0.21 | 71.49±0.27 | 77.68±0.20 |
| GNNSVD | 83.11±0.82 | 73.19±0.49 | 88.42±0.38 | 93.86±0.36 | 95.96±0.12 | 89.31±0.53 | 81.46±1.15 | 69.82±0.34 | 71.82±0.39 |
| ProGNN | 86.23±0.42 | 74.45±0.83 | 88.52±0.45 | (OOM) | (OOM) | 90.89±0.69 | 89.34±0.54 | (OOM) | (OOM) |
| NeuralSparse | 84.60±0.52 | 76.32±0.55 | 89.02±0.39 | 93.89±0.58 | 96.67±0.20 | 90.78±1.06 | 88.37±0.37 | (OOM) | (OOM) |
| PTDNet | 86.03±0.60 | 76.07±0.58 | 86.78±0.45 | 93.78±0.43 | 95.32±0.31 | 92.96±0.87 | 84.89±1.47 | (OOM) | (OOM) |
| CLNode | 86.60±0.64 | 77.23±0.54 | 88.76±0.57 | 94.13±0.34 | 96.87±0.45 | 93.90±0.42 | 89.57±0.62 | 71.54±0.20 | 78.40±0.41 |
| RCL | 86.90±0.39 | 78.95±0.18 | 90.14±0.43 | 95.05±0.23 | 96.88±0.19 | 95.06±0.52 | 90.47±0.38 | 73.13±0.14 | 79.89±0.35 |

**Initializing graph structure by a pre-trained model.** It is worth noting that the model needs an initial training graph structure $\mathbf{A}^{(0)}$ in the initial stage of training. An intuitive way is that we can initialize the model to work in a purely data-driven scenario that starts only with isolated nodes where no edges exist. However, an instructive initial structure can greatly reduce the search cost and computational burden. Inspired by many previous CL works [46, 13, 19, 55] that incorporate prior knowledge of a pre-trained model into designing curriculum for the current model, we initialize the training structure $\mathbf{A}^{(0)}$ by a pre-trained vanilla GNN model $f^*$. Specifically, we follow the same steps from line 4 to line 7 in the algorithm 1 to obtain the initial training structure $\mathbf{A}^{(0)}$ but the latent node embedding is extracted from the pre-trained model $f^*$.

**Implementation details.** We use the baseline model (GCN, GIN, GraphSAGE) as the backbone model for both our RCL method and all comparison methods. For a fair comparison, we require all models follow the same GNN architecture with two convolution layers. For each split, we run each model 10 times to reduce the variance in particular data splits. Test results are according to the best validation results. General training hyperparameters (such as learning rate or the number of training epochs) are equal for all models.

## 5.2 Effectiveness Results

Table 1 presents the node classification results of the synthetic datasets. We report the average accuracy and standard deviation for each model against the *homo* of generated graphs. From the table, we observe that our proposed method RCL consistently achieves the best or most competitive performance to all the comparison methods over three backbone GNN architectures. Specifically, RCL outperforms the second best method on average by 4.17%, 2.60%, and 1.06% on GCN, GIN, and GraphSAGE backbones, respectively. More importantly, the proposed RCL method performs significantly better than the second best model when the *homo* of generated graphs is low ($\leq 0.5$), on average by 6.55% on GCN, 4.17% on GIN, and 2.93% on GraphSAGE backbones. These demonstrate that our proposed RCL method significantly improves the model's capability of learning an effective representation to downstream tasks especially when the edge difficulties vary largely in the data.

We report the experimental results of the real-world datasets in Table 2. The results demonstrate the strength of our proposed method by consistently achieving the best results in all 9 datasets by GCN backbone architecture, all 9 datasets by GraphSAGE backbone architecture, and 8 out of 9 datasets by GIN backbone architecture. Specifically, our proposed method improved the performance of baseline models on average by 1.86%, 2.83%, and 1.62% over GCN, GIN, and GraphSAGE, and outperformed the second best models model on average by 1.37%, 2.49%, and 1.22% over the three backbone models, respectively. The results demonstrate that the proposed RCL method consistently improves the performance of GNN models in real-world scenarios.

Our experimental results are statically sound. In 43 out of 48 tasks our method outperforms the second-best performing model with strong statistical significance. Specifically, we have in 30 out of 43 cases with a significance $p < 0.001$, in 8 out of 43 cases with a significance $p < 0.01$, and in 5

Figure 2: Node classification accuracy (%) on Cora and Citeseer under random structure attack. The attack edge ratio is computed versus the original number of edges, where 100% means that the number of inserted edges is equal to the number of original edges.

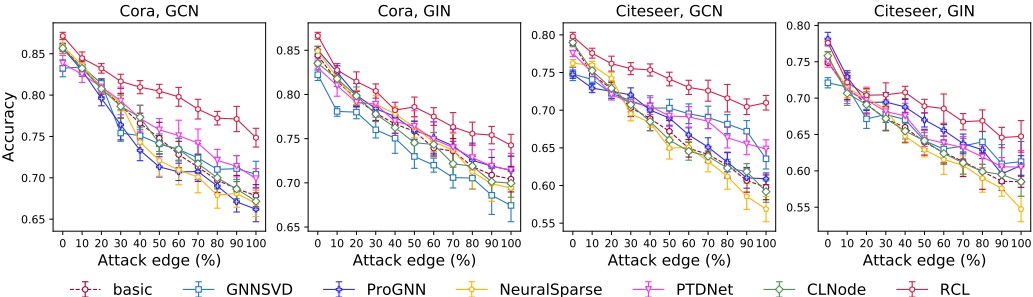

Table 3: Ablation study. Here "Full" represents the original method without removing any component. The best-performing method on each dataset is highlighted in bold.

|  | Synthetic1 | Synthetic2 | Citeseer | CS | Computers |
|---|---|---|---|---|---|
| Full | **73.98±0.55** | **97.42±0.17** | **79.79±0.55** | **94.66±0.22** | **90.23±0.23** |
| Curriculum-linear | 70.93±0.54 | 95.19±0.19 | 79.04±0.38 | 94.14±0.26 | 89.28±0.21 |
| Curriculum-root | 70.13±0.72 | 95.50±0.18 | 78.27±0.54 | 94.47±0.34 | 89.27±0.15 |
| Random-linear | 58.76±0.46 | 89.78±0.11 | 77.43±0.49 | 92.76±0.14 | 88.76±0.18 |
| Random-root | 61.04±0.20 | 91.04±0.09 | 76.81±0.35 | 92.92±0.15 | 88.81±0.28 |
| w/o edge appearance | 70.70±0.43 | 95.77±0.16 | 77.77±0.65 | 94.39±0.21 | 89.56±0.30 |
| w/o node confidence | 72.38±0.41 | 96.86±0.17 | 78.72±0.72 | 94.34±0.13 | 90.03±0.62 |
| w/o pre-trained model | 72.56±0.69 | 93.89±0.14 | 78.28±0.77 | 94.50±0.14 | 89.80±0.55 |

out of 43 cases with a significance $p < 0.05$. Such statistical significance results can demonstrate that our proposed method can consistently perform better than the baseline models in both scenarios.

## 5.3 Robustness Analysis Against Topological Noise

To further examine the robustness of the RCL method on extracting powerful representation from correlated data samples, we follow previous works [20, 31] to randomly inject fake edges into real-world graphs. This adversarial attack can be viewed as adding random noise to the topological structure of graphs. Specifically, we randomly connect $M$ pairs of previously unlinked nodes in the real-world datasets, where the value of $M$ varies from 10% to 100% of the original edges. We then train RCL and all the comparison methods on the attacked graph and evaluate the node classification performance. The results are shown in Figure 2, we can observe that RCL shows strong robustness to adversarial structural attacks by consistently outperforming all compared methods on all datasets. Especially, when the proportion of added noisy edges is large ($> 50\%$), the improvement becomes more significant. For instance, under the extremely noisy ratio at 100%, RCL outperforms the second best model by 4.43% and 2.83% on Cora dataset, and by 6.13%, 3.47% on Citeseer dataset, with GCN and GIN backbone models, respectively.

## 5.4 Ablation Study

To investigate the effectiveness of our proposed model with some simpler heuristics, we deploy a series of abalation analysis. We first train the model with node classification task purely and select the top K expected edges as suggested by the reviewer. Specifically, we follow previous works [43, 45] using two classical selection pacing functions as follows:

$$\text{Linear}\colon K_{\text{linear}}(t) = \frac{t}{T}|E|; \quad \text{Root}\colon K_{\text{root}}(t) = \sqrt{\frac{t}{T}}|E|,$$

where $t$ is the number of current iterations and $T$ is the number of total iterations, and $|E|$ is the number of total edges. We name these two variants Curriculum-linear and Curriculum-root, respectively. In addition, we also remove the edge difficulty measurement module and use random selection instead. Specifically, we gradually incorporate more edges into training in random order to verify the effectiveness of the learned curriculum. We name two variants as Random-linear and Random-root with the above two mentioned pacing functions, respectively.

In order to further investigate the impact of the proposed components of RCL. We also first consider variants of removing the edge smoothing components mentioned in Section 4.3. Specifically, we

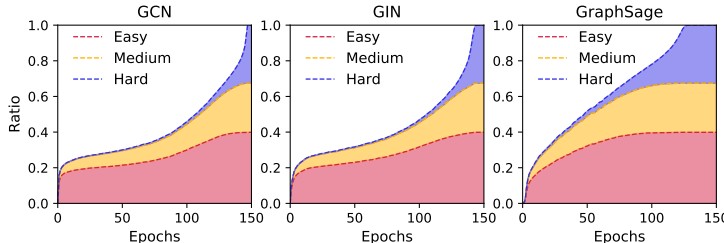
Figure 3: Visualization of edge selection process during training.

consider two variants *w/o EC* and *w/o NC*, which remove the smoothing function of the edge occurrence ratio and the component to reflect the degree of confidence for the latent node embedding in RCL, respectively. In addition to examining the effectiveness of edge smoothing components, we further consider a variant *w/o pre-trained model* that avoids using a pre-trained model to initialize model, which is mentioned in Section 5.1, to initialize the training structure by a pre-trained model and instead starts with inferred structure from isolated nodes with no connections.

We present the results of two synthetic datasets (*homophily coefficient*= $0.3, 0.6$) and three real-world datasets in Table 3. We summarize our findings from the above table as below: (i) Our full model consistently outperforms the two variants Curriculum-linear and Curriculum-root by an average of 1.59% on all datasets, suggesting that our pacing module can benefit model training. It is worth noting that these two variants also outperform the baseline vanilla GNN model Vanilla by an average of 1.92%, which supports the assumption that even a simple curriculum learning strategy can still improve model performance. (ii) We observe that the performance of the two variants Random-linear and Random-root on all datasets drops by 3.86% on average compared to the variants Curriculum-linear and Curriculum-root. Such behavior demonstrates the effectiveness of our proposed edge difficulty quantification module by showing that randomly involving edges into training cannot benefit model performance. (iii) We can observe a significant performance drop consistently for all variants that remove the structural smoothing techniques and initialization components. The results validate that all structural smoothing and initialization components can benefit the performance of RCL on the downstream tasks.

## 5.5 Visualization of Learned Edge Selection Curriculum

Besides the effectiveness and robustness of the RCL method on downstream classification results, it is also interesting to verify whether the learned edge selection curriculum satisfies the rule from easy to hard. Since real-world datasets do not have ground-truth labels of difficulty on edges, we conduct visualization experiments on synthetic datasets, where the difficulty of each edge can be indicated by its formation probability. Specifically, we classify edges into three balanced categories according to their difficulty: easy, medium, and hard. Here, we define all homogenous edges that connect nodes with the same class as easy, edges connecting nodes with adjacent classes as medium, and the remaining edges connecting nodes with far away classes as hard. We report the proportion of edges selected for each category during training in Figure 3. We can observe that RCL can effectively select most of the easy edges at the early stage of training, then more easy edges and most medium edges are gradually included during training, and most hard edges are left unselected until the end stage of training. Such edge selection behavior is highly consistent with the core idea of designing a curriculum for edge selection, which verifies that our proposed method can effectively design curriculums to select edges according to their difficulty from easy to hard.

## 6 Conclusion

This paper focuses on developing a novel CL method to improve the generalization ability and robustness of GNN models on learning representations of data samples with dependencies. The proposed method **R**elational **C**urriculum **L**earning (**RCL**) effectively addresses the unique challenges in designing CL strategy for handling dependencies. First, a self-supervised learning module is developed to select appropriate edges that are expected by the model. Then an optimization model is presented to iteratively increment the edges according to the model training status and a theoretical guarantee of the convergence on the optimization algorithm is given. Finally, an edge reweighting scheme is proposed to steady the numerical process by smoothing the training structure transition. Extensive experiments on synthetic and real-world datasets demonstrate the strength of RCL in improving the generalization ability and robustness.

## Acknowledgement

This work was supported by the National Science Foundation (NSF) Grant No. 1755850, No. 1841520, No. 2007716, No. 2007976, No. 1942594, No. 1907805, a Jeffress Memorial Trust Award, Amazon Research Award, NVIDIA GPU Grant, and Design Knowledge Company (subcontract number: 10827.002.120.04). The authors acknowledge Emory Computer Science department for providing computational resources and technical support that have contributed to the experimental results reported within this paper.

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

# A  Additional Experimental Settings and Results

## A.1  Additional Experimental Settings

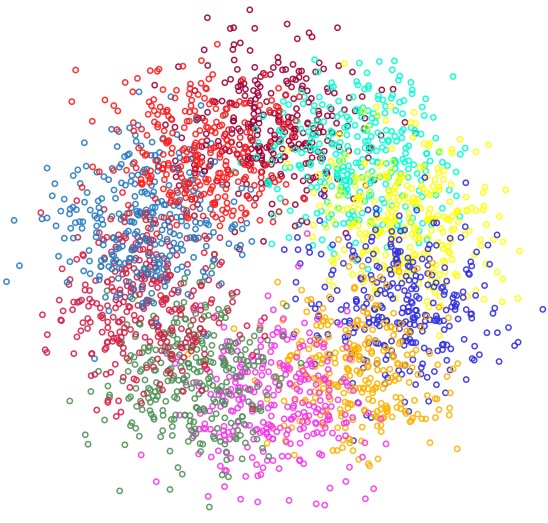

Figure 4: Visualization of synthetic datasets. Each color represents a class of nodes. Node attributes are sampled from overlapping multi-Gaussian distributions, where the attributes of nodes with close labels are likely to have short distances. Homogeneous edges represent edges that connect nodes of the same class (with the same color). The probability of connecting two nodes of different classes decreases with the distance between the center points of their class distribution. Therefore, the formation probability of a node denotes the edge difficulty, since edges between nodes with close classes are more likely to positively contribute to the prediction under the homogeneous assumption.

**Synthetic datasets.**  To evaluate the effectiveness of our proposed method on datasets with ground-truth difficulty labels on structure, we first follow previous studies [22, 1] to generate a set of synthetic datasets, where the difficulty of edges in generated graphs are indicated by their formation probability. Specifically, as shown in Figure 4, each generated graph is with 5,000 nodes, which are divided into 10 equally sized node classes $1, 2, \ldots, 10$. The node features are sampled from overlapping multi-Gaussian distributions. Each generated graph is associated with a *homophily coefficient (homo)* which indicates the likelihood of a node forming a connection to another node with the same label (same color in Figure 4). For example, a generated graph with *homo* $= 0.5$ will have on average half of the edges formed between nodes with the same label. For the rest edges that are formed between nodes with different labels (different colors in Figure 4), the probability of forming an edge is inversely proportional to the distances between their labels. Mathematically, the probability of forming an edge between node $u$ and node $v$ follows $p_{u \to v} \propto e^{-|c_u - c_v|}$, where the distances between labels $|c_u - c_v|$ means shortest distance of two classes on a circle. Therefore, the probability of forming an edge in the synthetic graph can reflect how well this edge is expected. Specifically, edges with a higher formation probability, e.g. connecting nodes with the same label or close labels, meaning that there is a higher chance that this connection will positively contribute to the prediction (less chance to be a noisy edge). Conversely, edges with a lower formation probability, e.g., connecting nodes with faraway labels, mean that there is a higher chance that this connection will negatively contribute to the prediction (higher chance to be a noisy edge). We vary the value of *homo* from $0.1, 0.2, \ldots, 0.9$ to generate nine graphs in total. Similar to previous works [22, 1], we randomly partition each synthetic graph into equal-sized train, validation, and test node splits.

**Implementation Details.**  We use the baseline model (GCN, GIN, GraphSage) as the backbone model for both our RCL method and all comparison methods. For a fair comparison, we require all models follow the same GNN architecture with two convolution layers. For each split, we run each model 10 times to reduce the variance in particular data splits. Test results are according to the best validation results. General training hyperparameters (such as learning rate or the number of training

epochs) are equal for all models. For the pre-trained model to initialize the training structure, we utilize the same model as the backbone model utilized by our method. For example, if we use GCN as the backbone model for RCL, the pre-trained model to initialize is also GCN. All experiments are conducted on a 64-bit machine with four NVIDIA Quadro RTX 8000 GPUs. The proposed method is implemented with Pytorch deep learning framework [32].

The following describes the details of our comparison models.

**Graph Neural Networks (GNNs).** We first introduce three baseline GNN models as follows.

**(i) GCN.** Graph Convolutional Networks (GCN) [23] is a commonly used GNN, which introduces a first-order approximation architecture of the Chebyshev spectral convolution operator;

**(ii) GIN.** Graph Isomorphism Networks (GIN) [50] is a variant of GNN, which has provably powerful discriminating power among the class of 1-order GNNs;

**(iii) GraphSage.** GraphSage [14] is a GNN method that computes the hidden representation of the root node by aggregating the hidden node representations hierarchically from bottom to top.

**Graph structure learning.** We then introduce four state-of-the-art methods for jointly learning the optimal graph structure and downstream tasks.

**(i) GNNSVD.** GNNSVD [9] first apply singular value decomposition (SVD) on the graph adjacency matrix to obtain a low-rank graph structure and apply GNN on the obtained low-rank structure;

**(ii) ProGNN.** ProGNN [20] is a method to defend against graph adversarial attacks by obtaining a sparse and low-rank graph structure from the input structure;

**(iii) NeuralSparse.** NeuralSparse [54] is a method to learn robust graph representations by iteratively sampling $k$-neighbor subgraphs for each node and sparsing the graph according to the performance on the node classification;

**(iv) PTDNet.** PTDNet [31] learns a sparsified graph by pruning task-irrelevant edges, where sparsity is controlled by regulating the number of edges.

**Curriculum learning on graph data.** We introduce a recent curriculum learning work on node classification as follows.

**(i) CLNode.** CLNode [45] regards nodes as data samples and gradually incorporates more nodes into training according to their difficulty. They apply a heuristic-based strategy to measure the difficulty of nodes, where the nodes that connect neighboring nodes with different classes are considered difficult.

**Searching space for hyperparameters.**
Number of epochs trained: $\{150, 500\}$;
Learning rate for model: $\{1e-2, 5e-3, 1e-3\}$;
Number of GNN layers: $\{2\}$;
Dimension of hidden state: $\{64\}$;
Age parameter $\lambda$ : $\{1, 2, 3, 4, 5\}$ (A larger value indicates faster pacing for adding edges, where 1 denotes the training structure will converge to the input structure at the final iteration).

## A.2 Additional Effectiveness Experiments on Heterophilic Datasets

| Dataset | Edge homo ratio | GCN | GCN-RCL | GIN | GIN-RCL |
|---|---|---|---|---|---|
| Texas | 0.11 | 0.5645 | **0.6006** | 0.5885 | **0.6156** |
| Cornell | 0.30 | 0.4084 | **0.5045** | 0.4234 | **0.4925** |
| Wisconsin | 0.21 | 0.4923 | **0.5294** | 0.5141 | **0.5599** |
| Actor | 0.22 | 0.2868 | **0.3186** | 0.2678 | **0.3006** |
| Squirrel | 0.22 | 0.2743 | **0.2999** | 0.2347 | **0.2519** |
| Chameleon | 0.23 | 0.3625 | **0.4385** | 0.3233 | **0.4033** |

Table 4: Node classfication results for six real-world heterophilic datasets, where the best performance of each model category in one dataset is highlighted.

In order to further verify the effectiveness of our proposed strategy on heterophilic graph datasets, we have included new experiments on six real-world heterophilic datasets. As shown in Table 4, our method consistently improve performance of backbone GNN models on these heterophilic datasets. Secifically, RCL outperforms the second best method on average by 5.04%, and 4.55%, on GCN and GIN backbones, respectively. The results can demonstrate our method is not limited to homophily graphs.

Although the inner product decoder utilized in experiments might imply an underlying homophily assumption, our method can still benefit from leveraging the edge curriculum present within the input datasets. A reasonable explanation is that standard GNN models are usually struggled with the heterophily edges, while our methodology designs a curriculum allowing more focus on homophily edges, which potentially leads to the observed performance boost.

## A.3 Additional Effectiveness Experiments on PNA Backbone Model.

| Dataset | PNA | PNA-RCL | PNA-linear | PNA-root | GCN | GCN-RCL | GCN-linear | GCN-root |
|---|---|---|---|---|---|---|---|---|
| Synthetic-0.3 | 0.6982 | **0.7667** | 0.7463 | 0.7445 | 0.6517 | **0.7398** | 0.6641 | 0.6533 |
| Synthetic-0.5 | 0.8742 | **0.9016** | 0.8476 | 0.8704 | 0.8715 | **0.9269** | 0.8494 | 0.8854 |
| Synthetic-0.7 | 0.9658 | **0.9821** | 0.9514 | 0.9766 | 0.9748 | **0.9962** | 0.9712 | 0.9796 |
| Cora | 0.8310 | **0.8521** | 0.8145 | 0.8254 | 0.8574 | **0.8715** | 0.8327 | 0.8553 |
| Citeseer | 0.7478 | **0.7652** | 0.7482 | 0.7505 | 0.7893 | **0.7979** | 0.7723 | 0.7814 |
| Computers | 0.8989 | **0.9096** | 0.8866 | 0.8975 | 0.8809 | **0.9023** | 0.8713 | 0.8985 |
| ogbn-arxiv | 0.7175 | **0.7441** | 0.6980 | 0.7242 | 0.7174 | **0.7408** | 0.7288 | 0.7359 |

Table 5: Node classfication results for our method and traditional CL methods using PNA and GCN as backbone. Here '-RCL' denotes our method, while '-linear' and '-root' denotes two traditional CL methods with different pacing functions.

In Table 5, new experiments that adopt modern GNN architecture - PNA model [6] have been added. From the table we can observe that our proposed method improves the performance of PNA backbone by 2.54% on average, which further verified the effectiveness of our method under different choices of backbone GNN model.

In addition, in Table 5 we further include two traditional CL methods for independent data as additional baselines, following classical works [2, 25]. We employed the supervised training loss of a pretrained GNN model as the difficulty metric, and selected two well-established pacing functions for curriculum design: linear and root pacing, defined as follows:

$$\text{Linear}: K_{\text{linear}}(t) = \frac{t}{T}|V|; \text{Root}: K_{\text{root}}(t) = \sqrt{\frac{t}{T}}|V|,$$

where $t$ is the number of current iterations and $T$ is the number of total iterations, and $|V|$ is the number of nodes.

We utilized GCN and PNA as backbone architectures, identified by the suffixes '-linear' and '-root'. Across all datasets, the results consistently demonstrate that our proposed method outperforms traditional CL approaches.

## A.4 Additional Robustness Experiments on PNA Backbone Model.

We present further robustness test against random noisy edges by using the PNA backbone model. The results are shown in Table 6, which further proves that our curriculum learning approach improves the robustness against edge noise with the advanced PNA model as the backbone.

## A.5 Time Complexity Analysis

Here we consider GCN as the backbone. First, the time complexity of an $L$-layer GCN is $O(L|\mathcal{E}|b + L|\mathcal{V}|b^2)$, where $b$ is the number of node attributes. Second, the time complexity of measuring the

| Dataset | Method | 0% | 10% | 20% | 30% | 40% | 50% | 60% | 70% | 80% | 90% |
|---------|--------|------|------|------|------|------|------|------|------|------|------|
| Cora | PNA | 0.8310 | 0.7911 | 0.7621 | 0.7402 | 0.7331 | 0.7210 | 0.6894 | 0.7042 | 0.6792 | 0.6617 |
| Cora | PNA-RCL | 0.8521 | 0.8315 | 0.8162 | 0.7969 | 0.7992 | 0.7951 | 0.7571 | 0.7642 | 0.7457 | 0.7371 |
| Citeseer | PNA | 0.7478 | 0.7195 | 0.7184 | 0.6934 | 0.6952 | 0.6920 | 0.6852 | 0.6552 | 0.6481 | 0.6327 |
| Citeseer | PNA-RCL | 0.7652 | 0.7422 | 0.7222 | 0.7254 | 0.7041 | 0.7012 | 0.6953 | 0.6921 | 0.6884 | 0.6794 |

Table 6: Further robustness test using PNA as backbone model. Here the percentage denotes the ratio of number of added random edges to the original edges.

| | Synthetic | Citeseer | Computers | ogbn-arxiv | ogbn-proteins |
|--------------|-----------|----------|-----------|------------|---------------|
| Vanilla | 7.32s | 3.90s | 16.88s | 55.22s | 1438.23s |
| GNNSVD | 11.49s | 3.82s | 35.96s | 135.72s | 2632.42s |
| CLNode | 6.29s | 3.96s | 17.02s | 58.53s | 1545.53s |
| ProGNN | 220.25s | 72.42s | 1953.23s | (-) | (-) |
| NeuralSparse | 310.02s | 88.91s | 6553.34s | (-) | (-) |
| PTDNet | 153.43s | 48.42s | 2942.02s | (-) | (-) |
| Ours | 4.07s | 2.42s | 14.62s | 71.49s | 2239.05s |

Table 7: Running time of our method and comparison methods. Here (-) denotes an out-of-memory error and Vanilla denotes the standard GNN model.

difficulty levels of edges by reconstruction is $O(|\mathcal{E}|d)$ where $d$ is the number of latent embedding dimensions. Third, the time complexity of selecting the edges to add is $O(|\mathcal{E}|)$. Therefore, the total time complexity of our algorithm is $O(|\mathcal{E}|(Lb + d) + L|\mathcal{V}|b^2)$.

In addition, we compare the total running time of our method and all comparison methods in the Table 7. We can observe that the running time of our proposed method is comparable to that of standard GNN models in all datasets. Notably, our method is even faster than standard GNN models in some datasets. One possible reason is that at the beginning of training, the graphs in our model have much fewer edges than those in standard GNN models. Therefore, the computational cost of the GNN model is also reduced.

## A.6 Parameter Sensitivity Analysis

Recall that RCL learns a curriculum to gradually add edges in a given input graph structure to the training process until all edges are included. An interesting question is how the speed of adding edges will affect the performance of the model. Here we conduct experiments to explore the impact of age parameter $\lambda$ which controls the speed of adding edges to the model performance. Here a larger value of $\lambda$ means that the training structure will converge to the input structure earlier. For example, $\lambda = 1$ means that the training structure will probably not converge to the input structure until the last iteration, and $\lambda = 5$ means that the training structure will converge to the input structure around half of the iterations are complete, and then the model will be trained with the full input structure for the remaining iterations. We present the results on two synthetic datasets (*homophily coefficient*$= 0.3, 0.6$) and two real-world datasets in Figure 5. As can be seen from the figure, the classification results are steady that the average standard deviation is only 0.41%. It is also worth noting that the peak values for all datasets consistently appear around $\lambda = 3$, which indicates that the best performance is when the training structure converges to the full input structure around two-thirds of the iterations are completed.

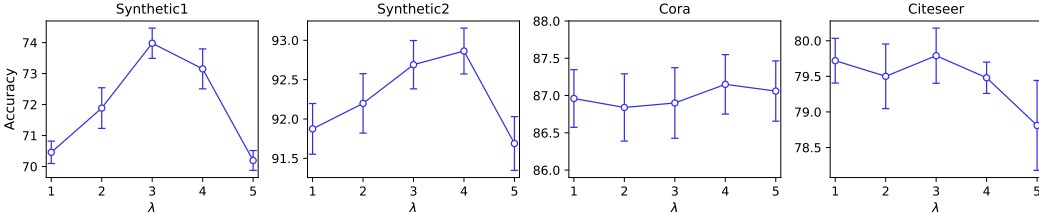

Figure 5: Parameter sensitivity analysis on four datasets. Here a larger value of $\lambda$ means the training structure will converge to the original structure at an earlier training stage.

## A.7 Visualization of Importance on Smoothing Component

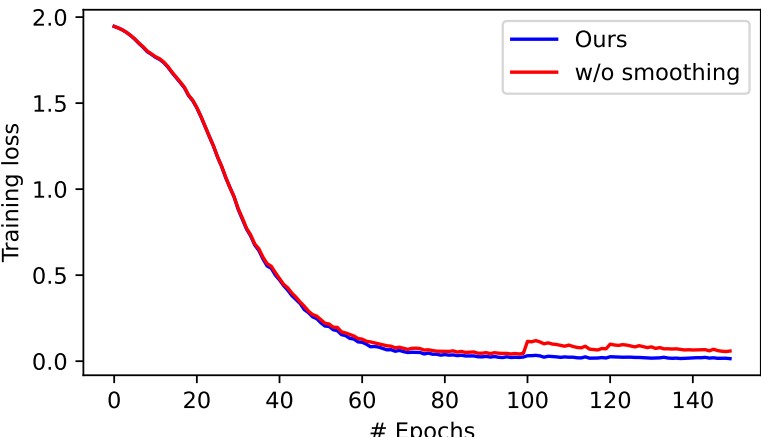

Figure 6: The comparison between our full model and the version without smoothing technique on the training loss trend.

Our experimental results demonstrated the importance of applying our smoothing component in stablizing the optimization process of training. Figure 6 shows that without the smoothing technique, the training loss spiked that reflects the GNN parameter shifts, which was caused by the number of edges discretely changed. However, after adding the smoothing technique, the training loss can smoothly converge, hence, the smoothing technique plays an important role in stabilizing the training process.

## B   Mathematical Proof

**Theorem 1.** We have the following convergence guarantees for Algorithm 1:
- **Avoidance of Saddle Points** If the second derivatives of $L(f(\mathbf{X}, \mathbf{A}^{(t)}; \mathbf{w}), \mathbf{y})$ and $g(\mathbf{S}; \lambda)$ are continuous, then for sufficiently large $\gamma$, any bounded sequence $(\mathbf{w}^{(t)}, \mathbf{S}^{(t)})$ generated by Algorithm 1 with random initializations will not converge to a strict saddle point of $F$ almost surely.
- **Second Order Convergence** If the second derivatives of $L(f(\mathbf{X}, \mathbf{A}^{(t)}; \mathbf{w}), \mathbf{y})$ and $g(\mathbf{S}; \lambda)$ are continuous, and $L(f(\mathbf{X}, \mathbf{A}^{(t)}; \mathbf{w}), \mathbf{y})$ and $g(\mathbf{S}; \lambda)$ satisfy the Kurdyka-Łojasiewicz (KL) property [40], then for sufficiently large $\gamma$, any bounded sequence $(\mathbf{w}^{(t)}, \mathbf{S}^{(t)})$ generated by Algorithm 1 with random initialization will almost surely converges to a second-order stationary point of $F$.

*Proof.* We prove this theorem by Theorem 10 and Corollary 3 from [27].
**[Avoidance of Saddle Points]** Because the sequence $(\mathbf{w}^{(t)}, \mathbf{S}^{(t)})$ is bounded, and the second derivatives of $L$ and $g$ are continuous, then they are bounded. In other words, we have $\max\{\|\nabla_{\mathbf{w}}^2 L(f(\mathbf{X}, \mathbf{A}^{(t)}; \mathbf{w}^{(t)}), \mathbf{y})\|, \|\nabla_{\mathbf{S}}^2 g(S^{(t)}; \lambda)\|\} \leq p$, where $p > 0$ is a constant. Similarly, it is easy to check that the second derivative of the term $\sum_{i,j} \mathbf{S}_{ij} \left\|\tilde{\mathbf{A}}_{ij}^{(t)} - \mathbf{A}_{ij}\right\|_2^2$ is bounded, i.e., $\max\{\left\|\nabla_{\mathbf{w}}^2 \sum_{i,j} \mathbf{S}_{ij} \left\|\tilde{\mathbf{A}}_{ij}^{(t)} - \mathbf{A}_{ij}\right\|_2^2\right\|, \left\|\nabla_{\mathbf{S}}^2 \sum_{i,j} \mathbf{S}_{ij} \left\|\tilde{\mathbf{A}}_{ij}^{(t)} - \mathbf{A}_{ij}\right\|_2^2\right\|\} \leq q$, where $q > 0$ is constant and $\tilde{\mathbf{A}}$ is a function of $\mathbf{w}$. Therefore, it means that the objective $F$ is bi-smooth, i.e. $\max\{\left\|\nabla_{\mathbf{w}}^2 F\right\|\}, \left\|\nabla_{\mathbf{S}}^2 F\right\|\} \leq p + q$. In other words, $F$ satisfies Assumption 4 from [27]. Moreover, the second derivative of $F$ is continuous. For any $\gamma > p + q$, any bounded sequence $(\mathbf{w}^{(t)}, \mathbf{S}^{(t)})$ generated by Algorithm 1 will not converge to a strict saddle of $F$ almost surely by Theorem 10 from [27].
**[Second Order Convergence]** From the above proof of avoidance of saddle points, we know that $F$ satisfies Assumption 4 from [27]. Moreover, because $L$ and $g$ satisfy the KL property, and the term $\sum_{i,j} \mathbf{S}_{ij} \left\|\tilde{\mathbf{A}}_{ij}^{(t)} - \mathbf{A}_{ij}\right\|_2^2$ satisfies the KL property, we conclude that $F$ satisfy the KL property as

well. From the proof above, we also know that the second derivative of $F$ is continuous. Because continuous differentiability implies Lipschitz continuity [47], it infers that the first derivative of $F$ is Lipschitz continuous. As a result, $F$ satisfies Assumption 1 from [27]. Because $F$ satisfies Assumptions 1 and 4, then for any $\gamma > p + q$, any bounded sequence $(\mathbf{w}^{(t)}, \mathbf{S}^{(t)})$ generated by Algorithm 1 will almost surely converges to a second-order stationary point of $F$ by Corollary 3 from [27]. □

While the convergence of Algorithm 1 entails the second-order optimality conditions of $f$ and $g$, some commonly used $f$ such as the GNN with sigmoid or tanh activations and some commonly used $g$ such as the squared $\ell_2$ norm satisfy the KL property [39, 40], and Algorithm 1 is guaranteed to avoid a strict saddle point and converges to a second-order stationary point.

