# OpenReview forum: "Curriculum Learning for Graph Neural Networks: Which Edges Should We Learn First"
_NeurIPS.cc/2023/Conference — NeurIPS 2023 poster_

### Official Review · Reviewer_4bWL · 2023-06-17

**Soundness:** 3 good
**Presentation:** 3 good
**Contribution:** 3 good
**Rating:** 6
**Confidence:** 4

**Summary:**

This paper proposes a novel curriculum learning method that progressively includes edges into training based on their difficulty level, starting from easy to hard. The difficulty level is determined by the expected model performance in predicting the edges.

**Strengths:**

(1) Graph representation learning is a very fundamental problem for graph-related problems, and designing curriculum learning strategy from edge perspective is a very interesting topic.

(2) The paper is well-organized and easy to be understood.

(3) Extensive experiments analysis is informative for the readers.

**Weaknesses:**

(1) Since the initial node embedding heavily relies on the quality of the encoder and the effectiveness of reconstruction loss, it is advantageous to initialize the training process using a pre-trained GNN encoder. Since the evaluation of edge difficulty in the paper is based on the difference between the reconstruction matrix and the ground-truth matrix, which leads to significant shift in the early stages of training from scratch. It is highly recommended to incorporate a pre-trained GNN encoder as an initialization step, which would be more reasonable.

(2) Since the main training strategy in the paper is curriculum learning, it is suggested to compare the impact of different pace functions on performance.

(3) As existing research suggests that curriculum learning is effective for noisy data, it would be valuable to validate the proposed method's effectiveness on different types and proportions of label rates.

(4) The paper lacks some recent literature on the latest advancements for graph neural networks or curriculum learning.
1. Curriculum Graph Machine Learning: A Survey. 2023
2. A Comprehensive Survey on Deep Graph Representation Learning. 2023
3. Graph Neural Network with Curriculum Learning for Imbalanced Node Classification. 2022

**Questions:**

Please address the aforementioned weakness.

**Limitations:**

No specific concern.

---

> ### Author Rebuttal · Authors · 2023-08-06
>
> Thank you for your valuable comments.
>
> Q1. 'Since the initial node embedding heavily relies on the quality of the encoder and the effectiveness of reconstruction loss, it is advantageous to initialize the training process using a pre-trained GNN encoder...'
>
> A1. Your suggestion aligns with the approach we have implemented in our experiments. As described in lines 297-305, a section titled 'Initializing graph structure by a pre-trained model' details our use of a pre-trained GNN to initialize edge difficulty measurement.
>
> Additionally, our ablation studies in Appendix Table 4 showed the performance difference when no pre-trained GNN model is employed. The results validate the beneficial impact of a pre-trained GNN on improving model performance by providing initial structure.
>
> Q2. 'Since the main training strategy in the paper is curriculum learning, it is suggested to compare the impact of different pace functions on performance.'
>
> A2. We indeed compared the impact of different pace functions on performance. In Appendix Table 4, we contrast our method with two commonly utilized pacing functions—linear and root. The results demonstrate that our designed pacing function can outperform these two competitive alternatives.
>
> Q3. 'As existing research suggests that curriculum learning is effective for noisy data, it would be valuable to validate the proposed method's effectiveness on different types and proportions of label rates.'
>
> A3. In Section 5.3 (refer to Figure 2), we conducted robustness tests against noisy edges, reflecting our primary focus on designing an appropriate curriculum on edges.
>
> We introduced noise into the graph by randomly incorporating noisy edges, ranging from 10\% to 100\% of the original edges. The findings reveal that our RCL model effectively alleviates the performance drop, reducing it by over 50\%. This demonstrates the robustness of our proposed curriculum learning technique when dealing with noisy scenarios.
>
>
> Q4. 'The paper lacks some recent literature on the latest advancements for graph neural networks or curriculum learning.'
>
> A4. We will include the recently related literature in our later revision.

---

> > ### Comment · Reviewer_4bWL · 2023-08-18
> >
> > The author's response partially addressed my concerns, and I will maintain my score.

---

### Official Review · Reviewer_tVbH · 2023-06-29

**Soundness:** 3 good
**Presentation:** 2 fair
**Contribution:** 3 good
**Rating:** 6
**Confidence:** 3

**Summary:**

The paper presents a curriculum learning strategy that works in the node classification setting. The key property in the node classification setting is that edges are not necessarily independent. The paper proposes a way to perform curriculum learning for this task, including the edges from easy to hard based on the distance of node embeddings (a model is trained on a subset of edges, then the easiest k edges are selected for the next round). The suggested method outperforms a number of GNNs in the area of graph structure learning which try to jointly learn the graph structure (removing noisy edges) and node embeddings. In particular, the suggested method is explicitly robust to random noise.

**Strengths:**

- a bottom-up curriculum strategy for node classification
- fully neuronal, not much of a heuristic (only the way the number of edges to add in the next round is some kind of a heuristic)
- The improvement in terms of robustness against noise is quite strong
- General improvement in terms of classification accuracies is nice (although modern and strong GNNs are missing both as comparison and backbone)
- adding the confidence of individual embeddings into the helps the model focussing on edges that are indeed "easy"

**Weaknesses:**

My major concern about the paper is that the experiments only use very basic algorithms (GCN, GIN, and GraphSAGE) as backbones for the proposed method. While I hope that the results carry over to modern GNNs, it would make sense to actually test that explicitly. My suggestion is to use PNA here as it is still a "standard" MPGNN without too fancy additions.

The second major criticism are the baseline methods used in the experiments. I believe that the task that is to be solved is still "just" node classification and thus all kinds of GNNs could serve as baseline, not only those that also try to remove noise. This holds for both the normal experiments as well as the experiment on robustness against noise (there is nothing adversarial about adding random edges). In addition, I believe that existing CL methods are worth comparing to, even if they assume an independence that definitely does not hold - the benefit of RCL should be even more pronounced for that case (and I would like to see that).

Further unsorted points I noted about the paper:
* the fairness criterion does not really feel "fair" to me - optimizing each method's hyperparameters individually would be preferred
 * it is unclear how the decoder is trained
 * Algorithm 1, Line 6 is very hard to read
 * I did not quite get how the number of edges in $A^0$ is determined
* It would be nice to note in the introduction that for graph learning (in contrast to node classification) the independence criterion of other CL methods holds. Just for the node classification task it is not applicable.
* It could be clearer what the difference to existing CL methods for node classification is - those are just mentioned to all perform "independent" CL even though targeting node classification
* 48: in how far is training unsupervised here? As far as I understood the model, the CL process is unsupervised but the overall training still needs supervision.
* 133: was the numerically stable process really a challenge in practice?
* To me the ablation study in the appendix was more interesting than Table 1. I believe that table 2 suffices to show the effectiveness of the method.
* Section 5.3: this is robustness against noise. There is no adversary here that selects the edges. It does not make the experiment less important. Here, an additional baseline using standard (modern) GNNs would be highly appreciated.
* The "adjacency" of classes sounded odd. Are the classes in the synthetic data really ordered and comparable? (i.e. 1 and 10 are further apart than 3 and 4)
* How strong/important is the given theorem? Is it surprising? Please add one more sentence about it other than that the proof is in the appendix. Is the convergence really that challenging?

Writing:
* please make sure that all figures are indeed vector graphics
* related work: please name the authors when using a paper as subject in a sentence
* The paragraph on initializing (the) graph structure by a pre-trained model exists both in the paper and the appendix.
* 134ff: this paragraph does not really add a lot of new information, nearly all of that was said before.
* Tables 1 and 2 should both have a very first column that indicates that the three blocks are the three backbones. (I would rotate the column by 90 degrees then it should still fit)
* Table 2: maybe exchange (-) by OOM to make clear what the problem was. Also specify that the available GPU memory was 48GB.
* Eq1: w is undefined and not used.
* I believe that mentioning the general speed of the method (with a pointer to the table in the appendix) would be benefitial.

Typos:
quite a number of "the" is missing, some appear in the wrong place.
* 82: are -> is
* 91: information that carried by the structure in data, thus -> information carried by the structure in the data, and thus
* Fig1 caption learn -> learns
* 103: potential -> potentially
* 112: represents -> represent (adding a second "let" would also be nice)
* 113: denotes there -> denotes that there
* 114: maps node feature matrix -> maps the node feature matrix
* 143: with theoretical -> with a theoretical
* 168 for node -> for the node
* 178: of K respect to -> of K with respect to
* Algo1, Input: a stepsize -> stepsize
* Algo1, Output: parameter w of GNN -> parameters w of the GNN
* 192ff: please use some tool like grammarly here, I marked 7 places in this paragraph
* 218: remove "mostly" (essentially only graph transformers do not only rely on message passing)
* 217: training -> intermediate
* 233: imposed to -> imposed on
* 248: shift -> differ
* 253: low confident -> low confidence
* 336: attck -> attack (but rather exchange the heading to Robustness analysis against structural noise - that would be a much more accurate heading in my opinion)

**Questions:**

- would it be possible to add experiments based on PNA? (or another modern backbone)

- would it be possible to add the other CL methods in the experiments? Even though they assume some independence which is typically not present?

- can an edge be included in iteration $t$ and not included in iteration $t+1$? Or are edges only added? (making this explicit in the paper would be very helpful to understand the edge selection better)

- How is the number of initial edges determined?

- Why is there no supervised loss on edges? (line 148) In Link prediction, one leaves out some percentage of the edges and tries to correctly predict them. This is clearly supervised. Why is such a scheme not applicable in this setting? (Or rather: its pretty close to the proposed strategy, would it make sense to add it?)

- Does Eq1 in combination with fractional edge weights mean that an edge weight of K is added in each iteration?

**Limitations:**

none.

---

> ### Author Rebuttal · Authors · 2023-08-06
>
> Thank you for your detailed comments and insightful suggestions.
>
> major 1. We present the results of PNA on synthetic and real-world datasets in global reply point 2 (Table 2 in PDF), which illustrate that our curriculum learning approach consistently improves the performance of PNA backbone by 2.54\% on average.
>
> major 2.
>
> New experiments. We used both GCN and PNA as backbone models for additional baselines, denoted by suffixes -linear' and -root'. The results are shown in global reply point 2 (Table 2 in PDF). The results consistently highlight that our proposed approach surpasses these conventional CL techniques across all datasets. Besides, it is also worth noting that these traditional CL methods can still enhance performance in most datasets (synthetic, Computers, and ogbn-arxiv), espcially with the Root pacing function.
>
> Choice of baselines. It is critical to note that our choice of baseline models is primarily guided by the attributes of our proposed CL strategy.
>
> First, our contribution is not a specific GNN model, but a general curriculum learning strategy that can improve the performance and robustness of all GNN models that follow a message passing mechanism. Our experimental results substantiate this claim, demonstrating that our CL strategy consistently improves performance across various GNN backbone models.
>
> Second, a unique aspect of our CL strategy is its dynamic alteration of the number of edges, placing it related to the domain of graph structure learning. Accordingly, we compared our method with four state-of-the-art graph structure learning methods.
>
>
>
>
> p1. We use validation set or cross-validation for fair comparison over all expriments. We require all models to follow the same key model architecture hyperparameters such as the number of graph convolution layers. We require all models to follow the same hyperparameter search space, which is indiciated in our Appendix lines 562-568.
>
>
> p2. As indicated in lines 171-172, our framework utilizes a non-parametric kernel function, e.g. the inner product kernel, as the decoder component. This ensures that there is no need for additional parameter training.
>
>
> p3. Algorithm 1, Line 6 describes the optimization step on the mask matrix $\mathbf{S}$, which results in more edges are gradually involved.
>
>
>
> p4. As described in line 297-305, we used a pretrained vanilla GNN model to help initialize the edges. Specifically, we use the pretrained model to extract the latent embeddings and our optimization model subsequently selects the `easiest' edges to form the initial edge set. The number of these selected edges is determined by our optimization model and parameter $\lambda$.
>
>
> p5. While we discussed the scenarios that motivated our method, to extend curriculum learning to handle data dependencies, we will ensure that we provide more clarity in the later revisions.
>
>
> p6. As we introduced in lines 87-93, existing CL methods for node classification generally treat nodes as independent samples during the learning process, thus can not well handle the correlation between data samples. These methods typically use heuristic metrics such as the node degree or loss as the indicator for designing the curriculum, which does not adequately address the dependencies between nodes.
>
> p7. The primary goal of our model remains supervised learning for the node classification task. The term `unsupervised' pertains to the curriculum designing process on the edges. Given that there are no readily computable supervised metrics on edges, we employ self-supervised task for the formulation of the curriculum on the edges.
>
>
> p8. Yes. In ablation studies Appendix Table 4 we have compared our full method with the version without smoothing technique, which can demonstrate the effectiveness of our smoothing techniques.
>
> Specifically, without edge smoothing, the training loss spiked when the number of edges discretely changed, which was caused by shifts in the optimal GNN parameters.
>
> p9. The lack of groundtruth difficulty levels for edges in real-world datasets motivated us to conducting extensive experiments on synthetic datasets, as represented in Table 1. Therefore, we constructed synthetic datasets with known ground-truth difficulty values. This design enables us to verify the capabilities of designed curriculum.
>
>
> p10. We present further robustness test against noise using PNA model in global reply point 3 (Table 3 in PDF).
> The results reveal that our curriculum learning approach remains robust against edge noise with the advanced PNA model as the backbone. Besides, we will change the title of this section into `Robustness analysis against topological noise' to better describe the experiments.
>
> p11. The classes in synthetic data are arranged in a circular order, which is visualized in Appendix Figure 4. This implies that classes 1 and 10 are equally separated with 3 and 4.
>
> p12. The theorem is important as it offers a robust theoretical basis for our approach's ability to handle the optimization challenges. As we mentioned above in p8, the discretely change of edges will result in the the spike of training loss, which is challenging for stable of the optimization process.
>
> q1. See major 1.
>
> q2. See major 2.
>
> q3. Is is very unlikely but possible an edge be included in iteration $t$ and not included in iteration $t+1$, which also verified the importance of adopting edge smoothing technique in ensuring the training stability.
>
> q4. See p4.
>
> q5. Our method operates in this manner, and we term it a self-supervised task, as the supervised learning focuses on node classification.
>
> q6. Our method can automatically determine the number of edges to be involved in each iteration, and it does not has to be a fixed constant.

---

> > ### Comment · Reviewer_tVbH · 2023-08-21
> >
> > I would like to apologize for the very late reply and thank the authors for their explanations and additional experiments. The rebuttal partly alleviated my concerns (especially the generalization part to other backbones seems to hold, generalizing from the "ancient" models used in the first iteration).
> >
> > P1 (fairness): I was mostly referring to the "two layer" restriction (line 307) which is suboptimal in many settings. Especially when one uses skip-connections (which should have become the default already some years back as it is never detrimental), deeper networks often perform much better. Even for a paper focusing on a new CL strategy, I believe that experiments in settings closer to SOTA models are helpful (and that includes using tricks like skip-connections and virtual nodes).
> >
> > Q3: it would be highly appreciated to add this comment to the main paper as this behavior is different from what has been described before on an intuitive level.
> >
> > Q6 follow-up: if K is not a constant but rather computed during runtime, how does the algorithm make sure that edges are indeed added? -> I found that in the response to another review, highlighting that $\lambda |S-A|$ is part of the optimization. And there is no guarantee that edges are added every round, it is just encouraged.
> >
> > As also highlighted by other reviewers, different pacing functions seem to be a key experiment that should not be hidden in the appendix.
> >
> > As pointed out by the first reviewer, the submission is harder to read than necessary. I believe that a revised manuscript could be much clearer and easier to follow. Furthermore, I would like to encourage additional time for proofreading for the camera ready version or next iteration of the paper. As a reader, I really want to think only about the method and not about all those grammar mistakes.
> >
> > Although I believe that content-wise, the paper has enough novelty for the conference  (and also experiments that support it) and would thus qualify as clear accept, I tend to keep my "weak accept" score, as the presentation should indeed be clearer. Even after reading the responses, I would not feel confident to reimplement the method in another project. For example, the function $g(S;\lambda)$ seems to be extremely important, but then its definition (not really longer) is somewhat hidden in line 185. Overall, the idea to use link prediction as a proxy to gauge how well edges are expected and then using those values to design a CL rule that adaptively includes more and more edges is not something that should be hard to describe.

---

### Official Review · Reviewer_f4A7 · 2023-07-20

**Soundness:** 3 good
**Presentation:** 3 good
**Contribution:** 3 good
**Rating:** 5
**Confidence:** 4

**Summary:**

This study addresses the challenge of varying learning difficulties among edges in a graph and proposes a curriculum learning approach that gradually incorporates more difficult edges. Experimental results on synthetic and real-world datasets demonstrate the effectiveness of the proposed method in improving accuracy and robustness.

**Strengths:**

1.	The research addresses an interesting and important problem of learning diverse difficulties among graph edges and understanding graph structures.
2.	The proposed method, which employs a self-supervised approach to measure edge difficulties, is novel. The motivation behind the curriculum learning method is well-founded.
3.	Significant accuracy improvements are observed on synthetic datasets across various settings. Figure 2 provides clear evidence of the method's effectiveness in enhancing robustness against noisy edges.


**Weaknesses:**

1.	The improvements on real-world datasets are not substantial.

**Questions:**

Could the inconsistent significance of the performance improvement in Table 2 be attributed to the edge selection method's limitations on these datasets? It appears that RCL performs better on large-scale datasets. Is there a specific reason for this observation?

---

> ### Author Rebuttal · Authors · 2023-08-06
>
> Thank you for your valuable comments.
>
> W1. `The improvements on real-world datasets are not substantial.'
>
> A1: We would like to highlight, in line 323-330 and line 331-335, that our model: (1) Tops performance in 26 out of 27 tasks across nine real-world datasets, signifying its effectiveness; (2) Shows consistent improvement over three different GNN backbone models, affirming its generalizability; (3) Produces statistically sound results, outperforming the second-best model in 43 out of 48 tasks with a significance of $p<0.05$, and in 38 out of these 43 cases, with a significance of $p<0.01$.
>
> Q1. `Could the inconsistent significance of the performance improvement in Table 2 be attributed to the edge selection method's limitations on these datasets? It appears that RCL performs better on large-scale datasets. Is there a specific reason for this observation?'
>
> A1. As we mentioned above, our model can consistently improve the performance of backbone GNN by leveraging the designed curriculum. Although there still exists difference on the performance improvements level on different datasets, it is extremely difficult to understand the reason behind it due to the lack of ground truth difficulty level of edges.
>
> However, a plausible explanation for the seemingly smaller absolute improvement on certain datasets (such as CS, Physics, and Photo) is that the underlying models for these datasets already yield a notably high prediction accuracy (exceeding 90\%). Consequently, there is less room for significant improvement.
> In contrast, for large-scale datasets, the base models tend to exhibit lower performance (around 70\%), thus providing more opportunities for our model to enhance their performance.

---

### Official Review · Reviewer_TBej · 2023-07-23

**Soundness:** 3 good
**Presentation:** 3 good
**Contribution:** 3 good
**Rating:** 5
**Confidence:** 4

**Summary:**

Summary
This work explores continual learning on data that is not independent, but has dependencies, such as graph edges. Three issues are raised when transferring continual learning techniques to learning on graphs: 1. there is no simple way to evaluate how easy/hard an edge is; 2. the curriculum should include a gradual way of involving more edges during training based on model performance; 3. as the GNN will observe different topologies, trainability issues might arise. The solutions consist of 1) using a self-supervised module to select K easiest edges, 2) proposing a new objective using the Lagrangian multiplier and 3) having a smooth transition between structures across iterations by having edge reweighing depend on edge selection occurrences. The continual learning scheme proposed is applied to standard GNNs such as GCN, GIN, GraphSAGE, showing improvements for most of them on synthetic and real-world node classification datasets.


**Strengths:**

I find that the paper is above the acceptance bar in its current form. In particular, the area is of interest to the graph community, the proposed approach is sensible, the writing is clear and all experiments were successful in the sense that the continual modifications to the adjacency improve the final performance over having the same GNN model use the given graph as input. The visualisation of the learnt edge curriculum also provides interesting insights.


**Weaknesses:**

My understanding is that, based on the fact that edges are selected based on similarity in embedding space, the proposed approach might struggle with heterophilic graphs. While I appreciate the synthetic experiments that emphasise the good performance at different levels of homophily, I would encourage the authors to also consider real-world heterophilic datasets, such as those proposed in [1] or [2].

[1] - Lim, Derek, et al. "Large scale learning on non-homophilous graphs: New benchmarks and strong simple methods." Advances in Neural Information Processing Systems 34 (2021): 20887-20902.
[2]. - Platonov, Oleg, et al. "A critical look at the evaluation of GNNs under heterophily: are we really making progress?." arXiv preprint arXiv:2302.11640 (2023).

**Questions:**

Please see the weaknesses section.

**Limitations:**

The authors only briefly mentioned limitations of the proposed approach, I encourage them to include a paragraph discussing them.

---

> ### Author Rebuttal · Authors · 2023-08-06
>
> We thank the reviewer for your valuable comments and acknowledgement of our work.
>
> Q1. `My understanding is that, based on the fact that edges are selected based on similarity in embedding space, the proposed approach might struggle with heterophilic graphs. While I appreciate the synthetic experiments that emphasise the good performance at different levels of homophily, I would encourage the authors to also consider real-world heterophilic datasets, such as those proposed in [1] or [2].'
>
> A1. Thank you for your suggestion and we have expanded the experiments on six real-world heterophilic datasets. The results are presented in global reply point 1 (see PDF Table 1), which indicate consistent improvements using our method over baseline GNN models on these heterophilic datasets. It is worth noting that the improvements in these datasets are even more significant than the homophily datasets used in our datasets. Specifically, RCL outperforms the second best method on average by 5.04\%, and 4.55\%, on GCN and GIN backbones, respectively.
>
> Although the inner product decoder we utilized might imply an underlying homophily assumption, our method appears to still benefit from leveraging the edge curriculum present within the input datasets. A reasonable explanation is that standard GNN models are usually struggled with the heterophily edges, while our methodology designs a curriculum allowing more focus on homophily edges, which potentially leads to the observed performance boost.
>
> In addition, we will include a paragraph to discuss the limitations of our work in future revisions.

---

> > ### Comment · Reviewer_TBej · 2023-08-20
> >
> > I’d like to thank the authors for their response. While I appreciate the effort to run RCL on heterophilic graphs, the datasets chosen have been found to have significant limitations in [2] that, once fixed, make the improvements over standard GNNs irrelevant.  This is the reason why I initially suggested running RCL on [1] or [2]. In this context and as I am not sure how reliable the experiments of RCL on these datasets are, I will maintain my score.
> >
> >
> > [1] - Lim, Derek, et al. "Large scale learning on non-homophilous graphs: New benchmarks and strong simple methods." Advances in Neural Information Processing Systems 34 (2021): 20887-20902.
> > [2]. - Platonov, Oleg, et al. "A critical look at the evaluation of GNNs under heterophily: are we really making progress?." arXiv preprint arXiv:2302.11640 (2023).

---

### Official Review · Reviewer_u5nt · 2023-07-25

**Soundness:** 2 fair
**Presentation:** 1 poor
**Contribution:** 2 fair
**Rating:** 3
**Confidence:** 3

**Summary:**

This paper proposes a curriculum learning (CL) method for graph neural networks on the node classification task. Existing CL strategies are mostly designed for indepedent data samples, and cannot trivially generalize to graphs that contain data dependencies. The proposed solution, termed as Relational Curriculum Learning (RCL), learns to select edges from easy to hard in each iteration. The overall idea is to formulate curriculum learning as an optimization problem w.r.t. a discrete mask over the edges. Since it is hard to optimize the discrete mask, the authors relax the goal and alternatively optimize the GNN parameters and the edge mask in an EM fashion. Tricks like edge reweighting are applied to stablize the change in the edge mask. Experiments on synthetic datasets and 9 real-world datasets verify the effectiveness of RCL, though the improvement on real-world datasets are marginal given that RCL requires a pre-trained vanilla GNN model for initialization.

**Strengths:**

- S1: This paper proposes a curriculum learning method tailored to the need of node representation learning. Unlike classical CL methods that learn to select easy samples, RCL uses all node labels but learns to select easy edges for GNN propagation.
- S2: This paper conducts experiments on both synthetic datasets and real-world datasets. It achieves consistent improvement on 3 backbone GNN models and 9 real-world datasets, though the improvement is marginal.

**Weaknesses:**

- W1: This paper is badly written and not very easy to follow. The title is not precise considering the proposed model. I would suggest changing it to “Edge Curriculum Learning for Node Classification with Graph Neural Networks”, as the model only applies to homogeneous graphs rather than multi-relational graphs. The 2nd paragraph in the intro doesn’t have a good undelying logic. For example, why traditional CL strategies are insufficient is not clarified. The challenges claimed in the 3rd paragraph are also very weak. To my understanding, the first challenge might be a real one for classical CL methods, but the other two challenges seem to be fabricated for the proposed model, not general to CL methods. The first 2 paragraphs of Sec. 4 just repeat two paragraphs from the introduction.
- W2: There are some math and concept errors in the core statement of the proposed model. Eqn. 2 is not a faithful usage of Lagrange multiplier (or KKT conditions since the constraint is an inequality), since the number of edges K is missing there and the adjacency matrix A is introduced from nowhere. The alternative optimization in Algorithm 1 and Line 192-205 is not proximal optimization. Proximal gradient is the method that optimizes a continuous surrogate with gradient and projects the solution back to a discrete space. Here you are just alterate the optimization of the GNN model and the continuous mask, which is more like the EM algorithm.
- W3: The proposed RCL model lacks a clear high-level idea that can educate the community, nor an excellent performance that can make this model an off-the-shelf tool. Given that RCL requires a pretrained GNN model and new hyperparameters, needs a relatively complicated process for optmization, and only achieves slightly better results on real-world datasets, I doubt its signficance to the community.

**Questions:**

- Q1: Line 12-15: How does the proposed model handle the data dependency issue? It is not explicitly mentioned.
- Q2: Line 30-42: Can you describe the basic idea of traditional CL strategies and why they are insufficient for graphs? Can you describe how the dependencies in graphs impose a challenge for traditional CL methods?
- Q3: Line 51-57: Why is it difficult to design an appropriate curriculum to gradually involve edges? Does the difficulty for convergence only hold for your method or any CL methods in general?
- Q4: Line 65-66: You introduced the optimization model as a solution to an appropriate learning pace. How does the optimization model result in autoamtically increasing the number K to involve more edges? Some logic or description is missing here.
- Q5: Line 163: Does $\odot$ mean element-wise multiplication here?
- Q6: Line 170: The constraint, plus the residual errors, guarantee that only the most K well-expected edges are selected, right?
- Q7: Line 171-172: Please cite VGAE[1] for this dot-product style link prediction design.
- Q8: Algorithm 1 Line 6: Is the argmin taken over continuous S space or binary S space? I presume it’s continuous space here.
- Q9: Line 197-198: What do you mean by “the proximal terms” here?
- Q10: How do you guarantee that the number of edges grow monotonically during the curriculum learning?
- Q11: Why is the performance on Cora, Citeseer, PubMed higher than literature? Did you use a split other than the original split in the GCN paper? The authors should clarify that.
- Typos:
    - Title: Graph Neural Network → Graph Neural Networks
    - Line 11: cannot be trivially generalized to → cannot trivially generalize to
    - Line 195: You may add a reference link to Algorithm 1.
    - Line 200: extracts → extract
    - Line 219: recursively → iteratively
    - Table 2: GraphSage → GraphSAGE
    - Sec. 5.3: attck → attack

[1] Kipf and Welling. Variational Graph Auto-Encoders. NIPS 2016 workshop.

**Limitations:**

The authors don’t discuss the limitations and societal impacts in the paper. I would suggest the authors adding one paragraph to discuss that. For example, some limitations could be that RCL requires a longer training time since it needs to first train a vanilla GNN model and then finetune it with curriculum learning.

---

> ### Author Rebuttal · Authors · 2023-08-06
>
> We thank the reviewer for the detailed assessment and valuable suggestions.
>
> R1: The term `Relational' in our title is intended to emphasize our research on integrating inter-node relationships into Curriculum Learning (CL) strategies for GNN models.
>
> `why traditional CL strategies are insufficient is not clarified'
>
> We have elaborated why traditional CL strategies are insufficient in lines 40-42. Traditional CL strategies can only deal with independent data samples like images, which neglects relationships and structures in dependent data. Please refer to the second point in the global reply for extended comparison experiments with traditional CL methods, providing further empirical evidence.
>
> `...challenges seem to be fabricated for the proposed model, not general to CL methods'
>
> The challenges are not fabricated solely for the proposed model but indeed are general to CL methods on handling graph edges.
>
> (1) Formulating a progressive, edge-inclusive curriculum is a fundamental challenge for all CL methods. Our ablation studies (refer to Appendix Table 4) reinforce this point by showing that various previous involving functions struggle with performance. The results reveal the importance of an appropriate curriculum on selecting edges.
>
> (2) The drift of optimal GNN parameters, discussed in lines 55-57, is a universal challenge arises from the necessity to modify the number of edges, which is not unique to our method. For example, graph structure learning models.
>
>
> R2: `Eqn. 2 is not a faithful usage of Lagrange multiplier...'
>
> We acknowledge the need for more clarity regarding the use of the Lagrange multiplier method in Eqn. 2.
> We note that the inequality $||\mathbf{S}||_1 \geq K$ in Eqn. 1 is equivalent to the equality $||\mathbf{S}||_1 = K$. This is because the second term $\beta \sum\_{i,j} \mathbf{S}\_{{ij}} \mathbf{R}\_{{ij}}$ in the loss function would always encourage fewer selected edges by the mask matrix $\mathbf{S}$, as all values in the residual error matrix $\mathbf{R}$ and mask matrix $\mathbf{S}$ are nonnegative. This aligns with our motivation discussed on line 163 of the paper, `To filter out the edges with $K$ smallest residual error'.
>
> Given this, we can incorporate the equality constraint as a Lagrange multiplier and rewrite the loss function as
>
> $\mathcal{L}= L_{{GNN}} + \beta \sum\_{i,j} \mathbf{S}\_{{ij}} \mathbf{R}\_{{ij}} - \lambda (||\mathbf{S}||_1 - K)$.
>
> As $K$ is a constant value, minimizing the loss function is equivalent to minimizing the Eqn. 2 in our paper:
>     $\min\limits\_{\mathbf{w}, \mathbf{S}}   L\_{{GNN}} + \beta \sum\_{i,j} \mathbf{S}\_{ij} \mathbf{R}\_{ij} + \lambda || \mathbf{S} - \mathbf{A} ||,$
>
> where $\mathbf{A}$ is the input adjancency matrix.
>
>
> `...is not proximal optimization.'
>
> We acknowledge that this is a misusage of terminology. `EM-style alternative optimization' is more appropriate in describing our method.
>
> It is worth noting that the functionality of our proposed method should remain valid, despite this misuse of terminology.
>
>
> R3.
> 1. Our method does not strictly require a pretrained GNN model. Refer to global reply point 4 for details.
>
> 2. Our proposed RCL methodology consistently enhances the performance of GNN models across a variety of GNN backbones with statistical soundness p>0.01 in 38 out of 48 tasks.
>
> 3. Addressing the high-level idea, the RCL model introduces a novel perspective on handling data dependencies with curriculum learning, which can stimulate further research in both curriculum learning and graph representation learning areas.
>
>
> (Belows are replies to questions)
>
> A1. In the lines 109-110, we described treating data samples as nodes and their dependencies as edges. We then devised a CL strategy that leverages the inherent difficulty level of the edges to enhance the performance of the GNN model.
>
>
> A2. As we discussed in lines 48-50 and 122-124, traditional CL strategies usually use supervised computable metrics (e.g. training loss) to first quantify sample difficulty, and then gradually incorporate edges from easy to hard during the training process. However, quantifying the difficulty level of edges where no supervision is available is challenging, since supervised tasks typically associated with nodes.
>
>
> A3. See reply in R1 above.
>
>
> A4. We elaborated in Section 4.2 (lines 185-191). The regularization term $g(\mathbf{S};\lambda) = \lambda ||\mathbf{S}-\mathbf{A}||$ in Eqn. 2 allows control over edge incrementation through parameter $\lambda$, which increases with the number of training epochs. As $\lambda$ grows, the term $g(\mathbf{S};\lambda)$ push the mask matrix $\mathbf{S}$ to gradually approach the input adjacency matrix $\mathbf{A}$, thus progressively involving more edges in training.
>
>
> A5. Yes, it means element-wise multiplication.
>
> A6. Yes, as we mentioned above in response to `W2', only the most $K$ well-expected edges will be selected.
>
> A7. We will add the citation in our later revision.
>
> A8. Yes, it is continuous space in Algorithm 1 Line 6.
>
> A9. We refer to the last term of Algorithm 1 Line 3 and Line 6.
>
> A10. As we illustrated in response A4, the discrepancy penalty between the mask matrix $\mathbf{S}$ and the input adjacency matrix intensifies as $\lambda$ increases. This ensures a progressive increase in edge involvement during learning, continuing until the number of selected edges equals the total input edges.
>
> A11. In line 285, we have clarified that we follow the data splits from previous study on these three datasets, which is a commonly used split (adopted by Pytorch-geometric library, refer to 'full' split).
>
> We commit to correcting all the typographical errors and will include a paragraph discussing the limitations of our work in our subsequent revision.

---

> > ### Comment · Reviewer_u5nt · 2023-08-17
> > **Discussion**
> >
> > Thanks the authors for their response. The authors addressed some of my conerns but the major concern about the contribution and significance remains.
> >
> > **W1**
> >
> > I understand that the authors chose the title to emphasize the selection of edges in their method. My concern is that the term "relational" usually refers to graphs with typed edges, e.g. knowledge graphs. Therefore, I feel edge curriculum learning might be more precise here.
> >
> > For the weakness of traditional CL methods, I know the authors tried to claim that in Line 40-42, but my original concern is that Line 40-42 are not well supported. It would be better if the authors can add a sentence to illustrate the high-level idea of traditional methods and why they fail on graph data.
> >
> > I am still not convinced that the last two of three challenges are not fabricated. Edge curriculum learning is the technique proposed in this paper, so challenge (2) is more like for the proposed technique, not general to CL methods on graphs. (3) is like a vague challenge that can seemingly fit any GNN (not necessary CL methods) and there isn't a good way to verify the proposed method really solved this challenge.
> >
> > **W2**
> >
> > The authors are correct. If one read Eqn. 1 alone, the only solution is to use KKT conditions. The authors should mention that the residual error $R_ij$ is non-negative, thereby converting the inequality to equality. Lagrange multipliers can only be applied to equality constraints.
> >
> > Thanks for acknowledging the terminology problem.
> >
> > **W3**
> >
> > If I understand correctly, the baseline method CLNode doesn't use pretrained GNNs. Therefore, I would suggest using RCL w/o pre-trained GNNs for a fair comparison. The improvement seems to be marginal compared to CLNode. **Given that baselines seem to be reproduced by the authors in a setting different to their original papers, I doubt whether such marginal improvement is real or not, as it may be caused by different levels of engineering efforts on baselines and the proposed RCL.** Other reviewers and AC may correct me if they can confirm these results are significant enough for the community.
> >
> > **Q11**
> >
> > Thanks for confirming the data splits. Since methods like CLNode is different from the settings in its original paper, please clarify which results you reproduced from the official code and which results you copied from the original paper.

---

> > > ### Author Response · Authors · 2023-08-18
> > > **Response to Discussion (1/2)**
> > >
> > > R1.``I understand that the authors chose the title to emphasize the selection of edges in their method. My concern is that the term "relational" usually refers to graphs with typed edges, e.g. knowledge graphs. Therefore, I feel edge curriculum learning might be more precise here.''
> > >
> > > Thank you for your suggestion. We will emphasize that our method is for designing curriculum for graph edges and avoid the term `relational' in the revised title.
> > >
> > > ``For the weakness of traditional CL methods, I know the authors tried to claim that in Line 40-42, but my original concern is that Line 40-42 are not well supported. It would be better if the authors can add a sentence to illustrate the high-level idea of traditional methods and why they fail on graph data.''
> > >
> > >
> > > We did not claim traditional CL methods fail on graph data. Instead, we argue that traditional CL methods are not designed to handle the curriculum of the dependencies between nodes in graph data, which are crucial. Traditional CL methods focus on nodes' curriculum according to the difficulty of predictions on individual nodes. However, we aim to learn the curriculum of edges which requires inferring the difficulty levels of edge predictions based on the information of nodes and their dependencies.
> > >
> > > In addition, experimental results demonstrated our method, which addresses edge curriculum, outperforms those that don't, as shown in Table 2 in global rebuttal PDF. Specifically,
> > > across all seven datasets, our method consistently outperformed comparison methods by 3.3\% on average.
> > >
> > >
> > >
> > > ``I am still not convinced that the last two of three challenges are not fabricated. Edge curriculum learning is the technique proposed in this paper, so challenge (2) is more like for the proposed technique, not general to CL methods on graphs. (3) is like a vague challenge that can seemingly fit any GNN (not necessary CL methods) and there isn't a good way to verify the proposed method really solved this challenge.''
> > >
> > > The challenge (2) is to emphasize the importance of designing a proper pacing function for CL methods on graph data. As existing CL methods for graph data typically use fixed pacing function to involve samples, they can not provide flexibility to adjust the learning pace that is optimal to the model training status. Designing an adaptive pacing function for handling graph data is difficult since it requires joint optimization of both supervised learning tasks on nodes and the number of chosen edges. Therefore, this challenge is not just about the edge curriculum but about the open problem of adaptive pacing of CL in graph data. We will highlight this in the revised version.
> > >
> > >
> > > Challenge (3) is of interest to the community of graph structure learning, which studies the joint optimization of graph neural network models and graph structures. Our experimental results demonstrated this is indeed an open problem in this community and our technique has effectiveness in solving it. We have produced a new figure shows that without the smoothing technique, the training loss spiked that reflects the GNN parameter shifts, which was caused by the number of edges discretely changed. However, after adding the smoothing technique, the training loss can smoothly converge, hence, the smoothing technique plays an important role in stabilizing the training process. We can not post link to figure as required by NeurIPS official comment rule but we will include the figure in the revised paper.

---

> > > > ### Author Response · Authors · 2023-08-18
> > > > **Response to Discussion (2/2)**
> > > >
> > > > R2. Thank you for acknowledging our rebuttal, we will include the explanations to paper for a clear process.
> > > >
> > > > R3. (1) Our proposed method shows consistent better performance than the CLNode method. As presented in our main results Table 1 and Table 2, our proposed method consistently outperformed CLNode in 47 out of 54 tasks across nine synthetic datasets and nine real-world datasets using three different GNN backbone models (GCN, GIN, and GraphSAGE). Notably, our method achieved on average 2.62\% performance improvement over the CLNode method, with statistically significant improvements ($p<0.01$) observed in 36 of these 47 tasks. The consistent performance gain over various datasets and backbone GNN models can demonstrate the efficacy of our proposed method compared to CLNode method.
> > > >
> > > > Homo ratio   | 0.1 | 0.2 | 0.3 | 0.4 | 0.5 | 0.6 | 0.7 | 0.8 | 0.9
> > > >
> > > > CLNodes-GCN | 50.37$\pm$0.73                | 56.64$\pm$0.56                | 65.04$\pm$0.66                | 77.52$\pm$0.48                | 86.85$\pm$0.44                | 93.10$\pm$0.47                | 97.34$\pm$0.25                | 99.02$\pm$0.18                | 99.88$\pm$0.04
> > > >
> > > > RCL-GCN | 57.57$\pm$0.43       | 62.06$\pm$0.28       | 73.98$\pm$0.55       | 84.54$\pm$0.75       | 92.69$\pm$0.09       | 97.42$\pm$0.17       | 99.62$\pm$0.05       | 99.89$\pm$0.02       | 99.93$\pm$0.06
> > > >
> > > > CLNodes-GIN | 45.36$\pm$1.42                | 51.10$\pm$1.15                | 62.53$\pm$0.88                | 75.83$\pm$1.07                | 87.76$\pm$0.90         | 94.25$\pm$0.44        | 98.30$\pm$0.26       | 99.60$\pm$0.09       | 99.92$\pm$0.03
> > > >
> > > > RCL-GIN | 57.63$\pm$0.66       | 62.08$\pm$1.17       | 71.02$\pm$0.61       | 80.61$\pm$0.69       | 88.62$\pm$0.43       | 94.88$\pm$0.36       | 98.19$\pm$0.19         | 99.32$\pm$0.08                | 99.89$\pm$0.04
> > > >
> > > > CLNodes-SAGE | 69.41$\pm$0.66       | 70.83$\pm$0.58          | 75.51$\pm$0.36          | 82.65$\pm$0.43          | 87.08$\pm$0.56          | 91.58$\pm$0.41                | 95.91$\pm$0.38                | 98.33$\pm$0.26          | 99.57$\pm$0.14
> > > >
> > > > RCL-SAGE  | 68.03$\pm$0.37          | 71.39$\pm$0.51       | 76.99$\pm$0.99       | 83.76$\pm$0.55       | 88.24$\pm$0.30       | 93.34$\pm$0.56       | 97.66$\pm$0.52       | 98.86$\pm$0.28       | 99.64$\pm$0.08
> > > >
> > > >
> > > > | Cora       | Citeseer   | Pubmed     | CS         | Physics    | Photo      | Computers  | ogbn-arxiv  | ogbn-proteins
> > > >
> > > > CLNode-GCN |  85.67$\pm$0.33   |   78.99$\pm$0.57   |  89.50$\pm$0.28   | 93.83$\pm$0.24          | 95.76$\pm$0.16  |  93.39$\pm$0.83   |  89.28$\pm$0.38  | 70.95$\pm$0.18  | 71.40$\pm$0.32
> > > >
> > > > RCL-GCN | 87.15$\pm$0.44 | 79.79$\pm$0.55 | 89.79$\pm$0.12 | 94.66$\pm$0.32 | 97.02$\pm$0.23 | 94.41$\pm$0.76 | 90.23$\pm$0.23 | 74.08$\pm$0.33   | 75.19$\pm$0.26
> > > >
> > > > CLNode-GIN | 83.52$\pm$0.77          | 75.82$\pm$0.58          | 86.92$\pm$0.61  |  91.71$\pm$0.41  |  95.75$\pm$0.46   | 92.78$\pm$0.90          | 85.93$\pm$1.53     | 70.58$\pm$0.17      | 73.97$\pm$0.31
> > > >
> > > > RCL-GIN       | 86.64$\pm$0.39 |  77.60$\pm$0.18  | 89.17$\pm$0.29 | 93.92$\pm$0.27 | 96.75$\pm$0.17 | 93.88$\pm$0.51 | 89.76$\pm$0.19     | 72.55$\pm$0.15    | 78.76$\pm$0.22
> > > >
> > > > CLNode-SAGE |  86.60$\pm$0.64  | 77.23$\pm$0.54          | 88.76$\pm$0.57          | 94.13$\pm$0.34  |  96.87$\pm$0.45   |  93.90$\pm$0.42    |  89.57$\pm$0.62   | 71.54$\pm$0.20     | 78.40$\pm$0.41
> > > >
> > > > RCL-SAGE | 86.90$\pm$0.39 | 78.95$\pm$0.18 | 90.14$\pm$0.43 | 95.05$\pm$0.23 | 96.88$\pm$0.19 | 95.06$\pm$0.52 | 90.47$\pm$0.38    | 73.13$\pm$0.14    | 79.89$\pm$0.35
> > > >
> > > > (2)  Our proposed method can still significantly outperform CLNode even without the pre-trained model. Specifically,  Table 4 in Appendix shows the results of our method with and without a pre-trained model. Even without a pretrained model, our method can still outperform CLNode on average by 1.75\%.
> > > >
> > > > |  Synthetic-0.3 | Synthetic-0.6 | Citeseer | CS  | Computers |
> > > >
> > > > Ours-full | 73.98$\pm$0.55 | 97.42$\pm$0.17 | 79.79$\pm$0.55 | 94.66$\pm$0.22 | 90.23$\pm$0.23 |
> > > >
> > > > Ours w/o pretrained | 72.56$\pm$0.69 | 93.89$\pm$0.14 | 78.28$\pm$0.77 | 94.50$\pm$0.14 | 89.80$\pm$0.55 |
> > > >
> > > > CLNode | 65.04$\pm$0.66 | 93.10$\pm$0.47 | 78.99$\pm$0.57 | 93.83$\pm$0.24 | 89.28$\pm$0.38 |
> > > >
> > > > (3) For fair comparison, we require all the methods to follow the same hyperparameter searching space in GNN architecture and optimization process. We run all the methods 10 times for each data split to reduce the randomness for robust results. Besides, we have open-sourced all the code for reproducing the results of both our method and comparison methods in the submitted supplementary material.
> > > >
> > > > Q11.
> > > >
> > > > Thank you for your suggestion. Given the diverse experimental settings such as different choice of data splits presented in the papers of the comparison methods, we reproduced the results for all models with the same experimental settings and hyperparameter searching space for a fair comparison.

---

> > > > > ### Author Response · Authors · 2023-08-22
> > > > > **Consideration Regarding Reviewer u5nt's Feedback**
> > > > >
> > > > > Dear AC,
> > > > >
> > > > > We are writing to address a particular feedback situation regarding our recent submission.
> > > > >
> > > > > First and foremost, we would like to express our sincere gratitude for organizing a comprehensive rebuttal section. It has indeed been instrumental in allowing valuable inputs from the reviewers, which in turn has substantially aided the enhancement of our paper. Moreover, it has given us the platform to address and clarify any concerns raised.
> > > > >
> > > > > We'd like to bring to your attention the feedback from reviewer u5nt. The reviewer mentioned in his/her first response that our clarification addressed some of his/her concerns. However, we have not received any further response from him/her, despite promptly and thoroughly addressing a subsequent question seeking clarification. Considering that u5nt's review carries the only negative score, we kindly request you to take this situation into account when evaluating our submission.
> > > > >
> > > > > Thank you for your kind attention and understanding in this matter, and thank you again for organizing such a productive rebuttal session!
> > > > >
> > > > > Sincerely, Authors

---

### Author Rebuttal · Authors · 2023-08-06

We sincerely thank all the reviewers for your efforts in providing critiques and suggestions to our work. We summarize the newly expanded experiments and frequent questions as below:

1. We have included new experiments on six real-world heterophilic datasets. As shown in PDF Table 1, our method consistently improve performance of backbone GNN models on these heterophilic datasets. Secifically, RCL outperforms the backbone GNN on average by 5.04\%, and 4.55\%, on GCN and GIN backbones, respectively. The results can demonstrate our method is not limited to homophily graphs.

2. In PDF Table 2, new experiments that adopt modern GNN architecture - PNA model [1] have been added. From the table we can observe that our proposed method improves the performance of PNA backbone by 2.54\% on average, which further verified the effectiveness of our method under different choices of backbone GNN model.

In addition, in Table 2 we further include two traditional CL methods for independent data as additional baselines, following classical works [2,3]. We employed the supervised training loss of a pretrained GNN model as the difficulty metric, and selected two well-established pacing functions for curriculum design: linear and root pacing, defined as follows:

$$\mathrm{Linear\colon } K_{\mathrm{linear}}(t) = \frac{t}{T}|V|;$$
$$\mathrm{Root\colon } K_{\mathrm{root}}(t) = \sqrt{\frac{t}{T}}|V|,$$
where $t$ is the number of current iterations and $T$ is the number of total iterations, and $|V|$ is the number of nodes.

We utilized GCN and PNA as backbone architectures, identified by the suffixes '-linear' and '-root'. Across all datasets, the results consistently demonstrate that our proposed method outperforms traditional CL approaches.

[1] Corso, Gabriele, et al. "Principal neighbourhood aggregation for graph nets." Advances in Neural Information Processing Systems 33 (2020): 13260-13271.

[2] Bengio, Yoshua, et al. "Curriculum learning." Proceedings of the 26th annual international conference on machine learning. 2009.

[3] Kumar, M., Benjamin Packer, and Daphne Koller. "Self-paced learning for latent variable models." Advances in neural information processing systems 23 (2010).


3. We present further robustness test against random noisy edges by using the PNA backbone model. The results are shown in PDF Table 3, which further proves that our curriculum learning approach improves the robustness against edge noise with the advanced PNA model as the backbone.

4. We would like to clarify that while utilizing a pre-trained GNN model can help initialize the edges set, because its capacity to provide meaningful node embeddings to guide designing curriculum. However, the pre-trained model is not a necessity by our method. Our technique can operate effectively without such a setting. Our ablation studies in Appendix Table 4 can confirm that even without the pre-trained model to initialize, the method can still establish meaningful curriculum for numerous datasets.

---

### Decision · Program_Chairs · 2023-09-21

**Decision:**

Accept (poster)

**Comment:**

This work makes a novel contribution to curriculum learning design, in a way that incrementally processes edges fed into a graph machine learning system. The idea is highly interesting and likely to be industrially impactful.

That being said, I find that there are definitely things that the authors could have executed better about this work and the rebuttal; for example, the clarity of work was somewhat lacking, and the heterophilic benchmarks chosen should have followed the advice of Reviewer TBej to make them more trustworthy. I really hope that the authors will find some time to follow this advice when revising their paper one more time. Further, concerns were raised about the significance of the provided experimental results.

In spite of the questionable improvements on certain datasets, one must however acknowledge the sheer scale of the experimentation the authors have done, and I also find that the paper has made a good chunk of convincing results. Further, while maybe the paper in its submitted form is not sufficient for NeurIPS acceptance, given the results provided during the rebuttal and the changes the authors promise to make, I do see a relatively minor amount of effort necessary to push it over the bar.

As such, I will recommend acceptance of the paper.